# Scalable and Adaptive Trust-Region Learning via Projection Convex Hull

**Hongyang Jia[1], Qingchun Hou[2*], Bojun Du[1], Xiao Cai[3], Ning Zhang[1*], Chongqing Kang[1]**

[1]Department of Electrical Engineering, Tsinghua University
[2]ZJU-UIUC Institute, Zhejiang University
[3]Department of Electrical and Electronic Engineering, The University of Hong Kong
`{jhy21, dubj24}@mails.tsinghua.edu.cn`
`{ningzhang, cqkang}@tsinghua.edu.cn`
`houqingchun@zju.edu.cn, caix@hku.hk`

## Abstract

Learning compact and reliable convex hulls from data is a fundamental yet challenging problem with broad applications in classification, constraint learning, and decision optimization. We propose Projection Convex Hull (PCH), a scalable framework for learning polyhedral trust regions in high-dimensional spaces. Starting from an exact MINLP formulation, we derive an unconstrained surrogate objective and show that, under suitable weight assignments, the optimal hyperplanes of the MINLP are recovered as stationary points of the surrogate. Building on this theoretical foundation, PCH adaptively constructs and refines hyperplanes by subregion partition, strategic weight assignment, and gradient-based updates, yielding convex hulls that tightly enclose the positive class while excluding negatives. The learned polyhedra can serve as geometric trust regions to enhance selective classification and constraint learning. Extensive experiments on synthetic and real-world datasets demonstrate that PCH achieves strong performance in accuracy, scalability, and model compactness, outperforming classical geometric algorithms and recent optimization-based approaches, especially in high-dimensional and large-scale settings. These results confirm the value of PCH as a theoretically grounded and practically effective framework for trust-region learning. Codes are available at `https://github.com/IDO-Lab/trust-region-pch`.

## 1 Introduction

Convex hulls provide a fundamental geometric representation for modeling data boundaries (Preparata & Shamos, 2012), with broad applications in binary classification (Astorino et al., 2021; 2023), constraint learning (Fajemisin et al., 2024; Maragno et al., 2025), and decision optimization (Shen et al., 2023). In particular, when used as trust regions, convex hulls guarantee the inclusion of all positive samples while excluding infeasible regions, thereby characterizing the reliable domain of a classification model or defining data-supported feasibility sets in prescriptive decision-making. Classical computational geometry algorithms, e.g., QuickHull (QH) and its variants (Barber et al., 1996; Tzeng & Owens, 2012; Goodrich & Kitagawa, 2024), can construct exact convex hulls in low dimensions, but their complexity grows exponentially with dimensionality, and their facet-based representations offer little flexibility in controlling the number or arrangement of hyperplanes. These limitations make them unsuitable for modern learning tasks where convex hulls must be compact, controllable, and scalable to serve as effective trust regions under label-based supervision.

Recently, data-driven approaches have been proposed to directly learn convex hulls from labeled data (Balestriero et al., 2022; Jia et al., 2024). Projection-based methods (López et al., 2011) construct decision boundaries from extreme-point representations, but rely on global geometric operations that do not scale to large datasets. Multi-hyperplane classifiers (Raviv et al., 2018) improve flexibility by combining multiple linear components, yet offer limited structural regularization and cannot control model complexity. Mixed-integer programming (MIP) formulations (Barbato et al., 2024)

---
*Corresponding Authors

explicitly constrain the number of hyperplanes, but their reliance on binary variables quickly becomes intractable in high dimensions. Across these directions, existing methods still fail to address three fundamental requirements for trust region modeling: **(i) scalability** to high-dimensional data, **(ii) boundary tightness** to align with positive class boundaries, and **(iii) structural adaptivity** to match data complexity. Moreover, most approaches remain heuristic in nature and lack theoretical foundations that guarantee reliable trust regions, which is crucial in safety-critical applications.

Motivated by these gaps, we ask: Can we learn a convex hull that tightly encloses the positive class while excluding as many negatives as possible, under a limited budget of supporting hyperplanes? To that end, our approach builds on the observation that complex nonlinear decision boundaries admit locally linear approximations, allowing each subregion to be modeled efficiently by a supporting hyperplane. Based on theoretical foundation and starting from a mixed-integer nonlinear program (MINLP) that characterizes the tightest hull, we derive an unconstrained surrogate objective enabling scalable gradient-based updates. After each update, redundant hyperplanes are pruned, whereas new ones are introduced to further refine the boundary and exclude residual negatives. In practice, the surrogate guides hyperplanes toward class boundaries during training, yielding boundary-tight hulls. Together, these ideas form a divide-and-conquer framework, termed Projection Convex Hull (PCH), which is scalable, adaptive, boundary-tight, and theoretically motivated. An overview of the framework is provided in Figure 1, and its behavior in 2D is illustrated in Figure 2.

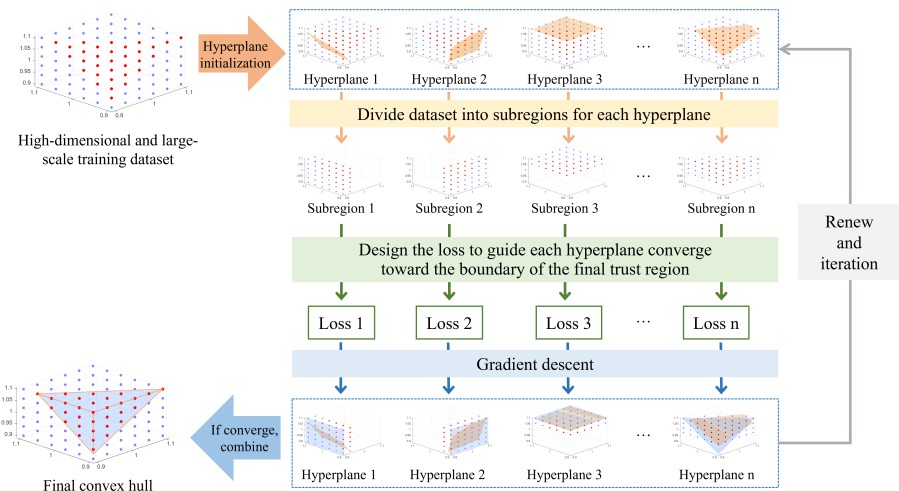

Figure 1: **Overview of the PCH framework.** At each iteration, data are partitioned into subregions assigned to hyperplanes, which are updated by gradient descent on the surrogate objective; hyperplanes are dynamically added or pruned, and the final convex hull is obtained by combining the active hyperplanes.

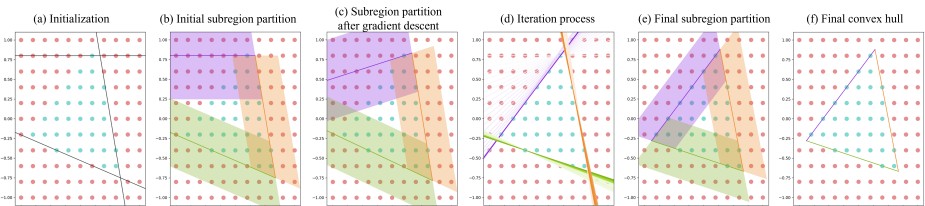

Figure 2: **Illustration of PCH training in 2D.** Hyperplanes are initialized and iteratively updated through iterative subregion refinement and surrogate optimization, yielding a compact convex hull.

Technically, we summarize our key contributions as follows:

1) **A principled formulation.** We establish a link between the intractable MINLP for tight convex hull learning and an unconstrained surrogate objective, providing a theoretically motivated foundation for scalable convex hull learning.

2) **A scalable divide-and-conquer framework.** We develop PCH, which iteratively refines subregions and their associated hyperplanes through gradient-based updates. The framework incorporates (i) partition-based subregion assignment, (ii) a surrogate objective to align hyperplanes with class boundaries, and (iii) adaptive hyperplane pruning and addition.

3) **Trust regions for learning and decision-making.** The learned convex hulls act as compact and reliable trust regions that can be embedded into classification models and constraint learning pipelines, enhancing both generalization and robustness.

4) **Extensive empirical validation.** Experiments on synthetic and real-world datasets show that PCH achieves high accuracy, superior scalability, and a significantly more compact structure compared to classical geometric algorithms and MIP-based baselines. Figure 3 presents key results on polyhedral datasets.

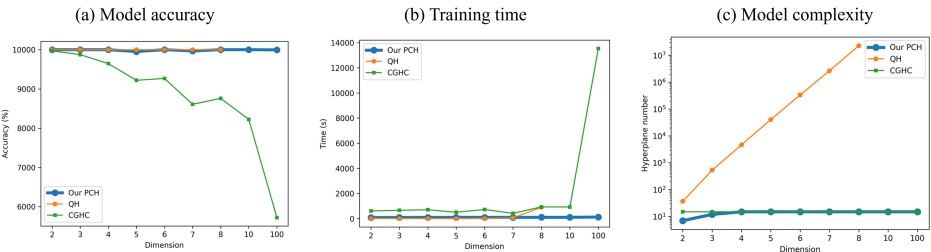

Figure 3: **PCH achieves consistent accuracy with superior scalability and compactness.** Across increasing dimensions, our method (PCH) maintains high classification accuracy while requiring significantly less training time and fewer hyperplanes than classical geometric and MIP-based baselines, i.e., QH from (Barber et al., 1996) and CGHC from (Barbato et al., 2024).

## 2 RELATED WORK

We review three lines of related work: classical convex hull construction, data-driven convex hull learning, and applications of convex approximations in machine learning and optimization.

**Classical geometric methods.** Traditional convex hull algorithms (Barber et al., 1996; Blelloch et al., 2020) compute exact hulls efficiently in low dimensions (Chan, 2011; Assarf et al., 2017; Gæde et al., 2024; Lv et al., 2025), with strong theoretical guarantees. However, the number of facets grows exponentially with dimension, making them impractical in high-dimensional or large-scale settings. Approximation methods such as DCH (Blum et al., 2019) alleviate this by selecting sparse representative subsets, but without label supervision cannot ensure inclusion of all positives. Moreover, classical methods produce vertex-based representations, where facet structure is determined implicitly, offering little control over hyperplane number or arrangement. These limitations motivate data-driven approaches.

**Data-driven convex hull learning.** Projection-based methods (López et al., 2011; Ding et al., 2017; Leng et al., 2019; Nemirko & Dulá, 2021) construct boundaries via nearest-point operations, but scale poorly with dataset size and dimension. Classifier-based polyhedral models, including multi-hyperplane learners (Yujian et al., 2010; Wang et al., 2011; Raviv et al., 2018; Leng et al., 2021) and SVM variants (Fan et al., 2008; Varol et al., 2017; Gu et al., 2018), improve flexibility and efficiency but offer little structural regularization or control over boundary tightness. Optimization-based formulations provide explicit structure: MIP methods (Zhu et al., 2020; Barbato et al., 2024; Phan et al., 2024) allow control over hyperplane number and arrangement, but their reliance on binary variables makes them intractable at scale. Our method addresses these gaps through a surrogate-based divide-and-conquer framework that reduces the global task to local hyperplane updates via gradient descent, avoiding binary enumeration.

**Applications in machine learning and optimization.** Convex hulls serve as trust regions in tasks such as selective classification (Ruano et al., 2015; Khosravani et al., 2016; Fajemisin et al., 2024) and constraint learning (Hou et al., 2020; Jia et al., 2024; Shi et al., 2024; Maragno et al., 2025; Jia et al., 2025). Recent neural methods, such as DeepHull (Balestriero et al., 2022; Liang et al., 2024) and Convex Polytope Tree (CPT) (Armandpour et al., 2021), approximate convex regions with deep networks, but lack explicit linear constraint form (e.g., $Ax \geq b$), hindering interpretability and complicating optimization embedding. Moreover, they do not enforce full inclusion of positives, leaving gaps in trust region coverage. Our method instead learns polyhedral hulls through surrogate-based optimization, yielding interpretable constraints, dynamic structural refinement, and boundary-tight trust regions with full positive coverage.

In summary, prior work has explored both geometric approximation and data-driven learning, but challenges remain in achieving scalability, adaptivity, and boundary tightness. Our framework addresses these by linking the exact MINLP formulation with a scalable surrogate, enabling gradient-based learning of convex hulls that are both compact and theoretically motivated.

## 3 PROBLEM FORMULATION

For a binary classification dataset $\mathcal{D} = \{(x_i, y_i)\}_{i=1}^n$ with $y_i \in \{+1, -1\}$, we model a trust region as a polyhedral convex region ("hull")

$$\mathcal{C} = \left\{ x \in \mathbb{R}^d \mid a^{s\top} x \geq b^s, \ s = 1, \ldots, S \right\}, \tag{1}$$

such that all positive samples $\mathcal{D}_+ = \{x_i : y_i = +1\}$ lie inside $\mathcal{C}$, while as many negatives $\mathcal{D}_- = \{x_i : y_i = -1\}$ as possible are excluded. We use $A = [a^1, \ldots, a^S] \in \mathbb{R}^{d \times S}$ and $b = [b^1, \ldots, b^S] \in \mathbb{R}^S$ to denote the collection of hyperplanes. All notations are summarized in Appendix A.

A standard formulation is a mixed-integer linear program (MILP; see Appendix B) that minimizes the number of enclosed negatives subject to full coverage of positives (Barbato et al., 2024). However, such MILP models primarily target negative exclusion and offer limited control over the *tightness* of the learned hull.

To encourage geometric tightness, we introduce margin variables and maximize the separation gap between negatives and the hull boundary via the following problem:

$$
\begin{aligned}
\max \quad & \textstyle\sum_s \xi^s \\
\text{s.t.} \quad & a^{s\top} x_i - b^s \ \geq \ 0, & \forall \, x_i \in \mathcal{D}_+, \ \forall s, \\
& a^{s\top} x_i - b^s \ \leq \ -\xi^s + \xi_i^s, & \forall \, x_i \in \mathcal{D}_-^{\mathrm{out}}, \ \forall s, \\
& z_i^s \xi_i^s = 0, \ z_i^s \in \{0, 1\}, \ \xi_i^s \geq 0, & \forall \, x_i \in \mathcal{D}_-^{\mathrm{out}}, \ \forall s, \\
& \textstyle\sum_s z_i^s \ \geq \ 1, & \forall \, x_i \in \mathcal{D}_-^{\mathrm{out}}, \\
& \|a^s\| = 1, \ \xi^s \geq 0, & \forall s,
\end{aligned}
\tag{2}
$$

where $\mathcal{D}_-^{\mathrm{out}} = \mathcal{D}_- \setminus \mathrm{conv}(\mathcal{D}_+)$ denotes negatives that lie outside the convex hull of positives. Each $(a^s, b^s)$ defines a hyperplane of $\mathcal{C}$. The variable $\xi^s$ is a nonnegative margin controlling the signed distance from hyperplane $s$ to the negatives it separates, and $\xi_i^s$ tracks violation for sample $x_i$ w.r.t. hyperplane $s$. The complementarity constraint $z_i^s \xi_i^s = 0$ enforces that a negative marked as separated by hyperplane $s$ ($z_i^s = 1$) incurs no violation ($\xi_i^s = 0$); otherwise a nonnegative slack is allowed. Requiring $\sum_s z_i^s \geq 1$ ensures that each $x_i \in \mathcal{D}_-^{\mathrm{out}}$ is separated by at least one hyperplane. Because of the binary-continuous complementarity and the normalization $\|a^s\| = 1$, Equation 2 is a MINLP problem.

The MINLP is computationally expensive in high dimensions and at large scale, which motivates replacing it with an *unconstrained surrogate objective* that preserves the essential geometric structure while enabling efficient gradient-based updates.

## 4 THEORETICAL FOUNDATION

Building on the MINLP formulation in Equation 2, we establish its connection to a family of unconstrained surrogate objectives by progressively decoupling it into hyperplane-wise subproblems. The goal of this section is threefold: (i) show that the intractable MINLP can be reduced to per-hyperplane separation problems, (ii) construct an unconstrained surrogate objective for gradient

descent, and (iii) characterize the weight assignment conditions under which stationary points of the surrogate recover the optimal hyperplanes of the MINLP. Proof of the theoretical foundation can be found in Appendix C.

**From MINLP to hyperplane subproblems.** For each supporting hyperplane $s = 1, \ldots, S$, let $\mathcal{D}_-^s := \{x_i \in \mathcal{D}_- \mid z_i^s = 1\}$ denote the subset of negatives separated by hyperplane $s$ in the optimal MINLP solution. Then each hyperplane $(a^s, b^s)$ solves the following constrained separation problem:

$$\begin{aligned}
\max \quad & \xi^s \\
\text{s.t.} \quad & a^{s\top} x_i - b^s \geq 0 && \forall x_i \in \mathcal{D}_+, \\
& a^{s\top} x_i - b^s \leq -\xi^s && \forall x_i \in \mathcal{D}_-^s, \\
& \|a^s\| = 1,
\end{aligned} \tag{3}$$

which seeks the tightest separating hyperplane for $\mathcal{D}_-^s$ while ensuring inclusion of all positives.

**Lemma 1** (Decomposition). *Let $(A^\star, b^\star)$ be the optimal solution to Equation 2. Then each $(a^{s\star}, b^{s\star})$ is an optimal solution to the corresponding subproblem Equation 3.*

**From constrained subproblem to unconstrained surrogate.** To obtain an unconstrained surrogate, we adopt a soft-split loss on the local dataset $\mathcal{D}^s := \mathcal{D}_+ \cup \mathcal{D}_-^s$. For a candidate hyperplane $(a, b)$ and sample $x_i \in \mathcal{D}^s$, we define the surrogate objective as:

$$\begin{aligned}
\min_{(a,b)} \quad & f(a,b) = -\frac{1}{2} \frac{\left(\sum \omega_i p_i^{\mathrm{L}} y_i\right)^2}{\sum \omega_i p_i^{\mathrm{L}}} - \frac{1}{2} \frac{\left(\sum \omega_i p_i^{\mathrm{R}} y_i\right)^2}{\sum \omega_i p_i^{\mathrm{R}}} \\
& p_i^{\mathrm{L}} = 1 / \left(1 + e^{\beta(a^\top x_i - b)/\|a\|}\right) \quad p_i^{\mathrm{R}} = 1 / \left(1 + e^{\beta(b - a^\top x_i)/\|a\|}\right) \quad \forall x_i \in \mathcal{D}^s
\end{aligned} \tag{4}$$

where $\beta > 0$ controls the sharpness of the soft split. $\omega_i \geq 0$ denote a weight assigned to $x_i$.

**Lemma 2** (Surrogate equivalence). *Let $(a^\star, b^\star)$ solve the constrained problem Equation 3. Then there exists a weight assignment $\{\omega_i\}$ such that $(a^\star, b^\star)$ is a stationary point of the surrogate objective Equation 4.*

*Intuition.* At a high level, the stationarity conditions of the surrogate Equation 4 require both $\partial f / \partial b = 0$ and $\partial f / \partial a = 0$. Linear constraints such as

$$\sum \omega_i = 2, \quad \sum \omega_i y_i = 0, \quad \sum \omega_i p_i^{\mathrm{R}} = 1, \quad \sum \omega_i p_i^{\mathrm{L}} p_i^{\mathrm{R}} y_i = 0, \tag{5}$$

can be seen as enforcing scale and class-balance moments, which ensure the $b$-direction stationarity condition at $b^\star(a) := \min_{x \in \mathcal{D}_+} a^\top x$. In contrast, the $a$-direction condition is more involved and can be approximated via the fixed-point relation in Lemma 3. Together, these constraints illustrate how suitable weight assignments allow the surrogate to mimic the objective of maximizing the separation gap in Equation 3.

**Stationarity and weight assignment.** The surrogate admits a fixed-point characterization:

$$a = \sum \omega_i \, p_i^{\mathrm{L}} p_i^{\mathrm{R}} \, y_i \, x_i, \tag{6}$$

which coincides with the stationarity condition of $f(a, b)$ once the weights satisfy the linear constraints in Equation 5. Since $p_i^{\mathrm{L}}$ and $p_i^{\mathrm{R}}$ depend on $a$, Equation 6 induces a complex coupling.

**Lemma 3** (Sufficient fixed-point condition). *If $(a, b)$ and nonnegative weights $\{\omega_i\}_{i \in \mathcal{D}^s}$ satisfy the fixed-point relation Equation 6 together with the linear constraints Equation 5, then $(a, b)$ is a stationary point of the surrogate objective Equation 4.*

**Theorem 1** (Main result). *Let $(A^\star, b^\star)$ be the optimal solution to the MINLP Equation 2. Then for each hyperplane $(a^{s\star}, b^{s\star})$, there exists a weight assignment $\{\omega_i^s\}$ such that $(a^{s\star}, b^{s\star})$ is a stationary point of the surrogate objective Equation 4.*

*Proof sketch.* Lemma 1 shows that the MINLP decomposes into hyperplane-wise subproblems. Lemma 2 establishes that the solution to each subproblem can be realized as a stationary point of the surrogate objective under suitable weights. Normalized weights satisfying Equation 5 ensure stationarity in the $b$-direction. Lemma 3 further provides a sufficient condition that guarantees stationarity in the $a$-direction via the fixed-point relation. Combining these results yields the claim.

These results justify the surrogate-based divide-and-conquer framework described next, where suitable weight assignment strategies allow us to implement the theoretical insights in practice. It is important to note that these theoretical results characterize the structural conditions that any MINLP optimum must satisfy. The surrogate leverages these necessary conditions to guide the hyperplane updates and weight assignment toward geometrically consistent stationary configurations.

## 5 METHODOLOGY

Our PCH framework (Figure 1) is motivated by the observation that complex and nonlinear classification boundaries often exhibit regional linearity and can be locally approximated by hyperplanes with small classification error. Identifying the locations of these regional boundaries is straightforward, and fitting them with linear hyperplanes is computationally efficient. By iteratively optimizing such region-specific hyperplanes and combining them, we construct a convex hull that is scalable, structurally adaptive, and geometrically tight. Specifically, the PCH framework operates as follows:

---
**Algorithm 1** Projection Convex Hull (PCH)

---
**Require:** Labeled dataset $\mathcal{D} = \{(x_i, y_i)\}$, hyperplane budget $S$
**Ensure:** Compact convex hull $(A, b)$ enclosing $\mathcal{D}_+$
 1: Initialize hyperplanes $\{a^s\}_{s=1}^S$ using PinCHD
 2: **while** not converged **do**
 3:    Place each hyperplane $s$ at $b^s = b^\star(a^s) = \min_{x \in \mathcal{D}_+} a^{s\top} x$
 4:    **Subregion assignment:** Build local subregions $\mathcal{D}^s$ by Equation 7.
 5:    **Strategic weight assignment:** Update weights by a gradient step $\omega \leftarrow \omega + \eta \frac{\partial \xi(\omega)}{\partial \omega}$ followed by projection $P_\mathcal{F}(\omega)$ in Equation 11
 6:    Apply one-step gradient descent on the surrogate objective Equation 4 to update $a^s$
 7:    **Adaptive structure adjust:** Prune redundant hyperplanes, and introduce new ones to further refine the boundary
 8: **end while**
 9: **return** Final convex hull $(A, b)$

---

### 5.1 SUBREGION ASSIGNMENT

For each hyperplane $s$, we build a local working set $\mathcal{D}^s$ on which the surrogate and the weight assignment are optimized. This realizes the hyperplane-wise decoupling suggested by Lemma 1 without solving the MINLP.

Given a normal $a$, we place the hyperplane at the positive support through $b^\star(a) = \min_{x \in \mathcal{D}_+} a^\top x$, so that the hyperplane touches the convex hull of positives. This choice aligns the surrogate update with boundary tightness. Then, we form a banded neighborhood around the hyperplane and enforce feasibility with respect to the other active hyperplanes:

$$\mathcal{D}^s = \left\{ x \in \mathcal{D} \,\Big|\, \underline{m}^s \le a^{s\top} x - b^\star(a^s) \le \overline{m}^s, \ A^{-s\top} x \ge b^{-s} \right\}, \tag{7}$$

where $A^{-s}$ and $b^{-s}$ stack all other hyperplanes except hyperplane $s$. The bandwidths $\underline{m}^s$ and $\overline{m}^s$ are chosen adaptively to include the most informative points and to ensure that the projected weight assignment can satisfy the linear constraints in Equation 5.

To guarantee that $\mathcal{D}^s$ contains sufficiently informative samples, we require it to include the top $d$ positive points with the largest split mistake $y_i(b^\star(a) - a^\top x_i)$. Accordingly, we set $\overline{m}^s$ using order statistics as

$$\overline{m}^s = \text{partition}_{(d)}\big(\{a^{s\top} x - b^\star(a^s) : x \in \mathcal{D}_+\}\big) + m_{\text{shift}}, \tag{8}$$

where $\text{partition}_{(d)}(\cdot)$ denotes the $d$-th order statistic (i.e., the $d$-th smallest value), ensuring that at least $d$ positives are included. The scalar $m_{\text{shift}}$ is a tunable hyperparameter that controls the bias of $\overline{m}^s$. Symmetrically, we obtain $\underline{m}^s$ using the same idea. In practice, the absolute value of $\underline{m}^s$ is typically smaller than $\overline{m}^s$, and we use the symmetric interval $[-\overline{m}^s, \overline{m}^s]$ in Equation 7. This construction improves numerical stability, ensures feasibility of the weight assignment constraints Equation 5, and empirically avoids degenerate configurations.

## 5.2 Strategic weight assignment

Having established in Lemma 2 and Lemma 3 that suitable weights enforce $b$-direction stationarity and guide $a$ toward the optimum of Equation 3, we now formulate a principled weight assignment strategy. The goal is to ensure that the surrogate objective Equation 4 remains closely aligned with the constrained separation problem Equation 3.

For each subregion $\mathcal{D}^s$, we assign weights $\omega = [\omega_i]_{x_i \in \mathcal{D}^s}$ by maximizing the separation gap, defined as the signed margin between the supporting positive and negative samples along the current $a(\omega)$:

$$\max_{\omega} \quad \xi(\omega) = \frac{\min_{x_i \in \mathcal{D}_+} a(\omega)^\top x_i - \max_{x_i \in \mathcal{D}_-^s} a(\omega)^\top x_i}{\|a(\omega)\|} \tag{9}$$
$$\text{s.t.} \quad C\omega = d, \quad \omega \in K_{\pi^+}, \quad \omega \in K_{\pi^-}, \quad \omega \geq 0,$$

where $K_{\pi^+}$ and $K_{\pi^-}$ are isotonic cones that enforce monotonicity of weights within the positive and negative subsets, respectively. Here $\pi^+$ and $\pi^-$ represent indices sorted by signed distance $a^\top x - b^\star(a)$. These constraints ensure that samples closer to the boundary are assigned larger weights, emphasizing the role of support vectors. The linear constraints $C\omega = d$ (encoding the four scalar equalities in Equation 5) guarantee $b$-direction stationarity, while maximizing $\xi(\omega)$ aligns the weight assignment with the objective of Equation 3.

Lemma 3 gives a fixed-point relation $a = g(a, \omega)$. Approximating $\partial a / \partial \omega \approx \partial g / \partial \omega$ enables a chain rule for the gradient of the separation gap:

$$\frac{\partial \xi(\omega)}{\partial \omega_i} = \frac{\partial \xi(\omega)}{\partial a} \frac{\partial a}{\partial \omega_i} = \frac{p_i^\mathrm{L} p_i^\mathrm{R} y_i}{\|\sum_j \omega_j\, p_j^\mathrm{L} p_j^\mathrm{R}\, y_j\, x_j\|} \left( x_i^\top (x_+ - x_-) - \frac{(a^\top(x_+ - x_-))(a^\top x_i)}{\|a\|^2} \right), \tag{10}$$

where $x_+ = \arg\min_{x \in \mathcal{D}_+} a^\top x$ and $x_- = \arg\max_{x \in \mathcal{D}_-^s} a^\top x$.

After a gradient update $\omega \leftarrow \omega + \eta \frac{\partial \xi(\omega)}{\partial \omega}$, with gradient update rate $\eta$, the weights may violate the constraints in Equation 9. We restore feasibility through successive projections:

$$P_\mathcal{F}(\omega) = P_S\big(P_{K_{\pi^+}}(P_{K_{\pi^-}}(P_+(\omega)))\big), \quad \mathcal{F} = S \cap \mathbb{R}_+^n \cap K_{\pi^+} \cap K_{\pi^-}, \tag{11}$$

where $P_S(\omega) = \omega - C^\top(CC^\top)^{-1}(C\omega - d)$ enforces the linear constraints in Equation 5, $P_+(\omega) = \max(\omega, 0)$ ensures nonnegativity, and $P_{K_{\pi^+}}, P_{K_{\pi^-}}$ are isotonic regression operators (pool-adjacent-violators algorithm, PAVA (De Leeuw et al., 2010)).

This strategic assignment ensures that $\omega$ dynamically guides $a$ toward maximizing the separation gap, thereby aligning surrogate optimization with the constrained objective Equation 3 and yielding boundary-tight updates in practice. Feasibility of the proposed projection method is discussed in Appendix D.

## 5.3 Adaptive structure adjust

To maintain compactness while preserving boundary tightness, PCH dynamically adjusts the number of active hyperplanes. **Addition:** if the current hull still encloses negative samples and the hyperplane budget is not exceeded, we add a new hyperplane initialized via a variant of the PinCHD algorithm (Leng et al., 2019), with a robust negative-sample selection strategy (details in Appendix E). **Pruning:** a hyperplane $s$ is removed once all samples in its subregion $\mathcal{D}^s$ lie on the nonnegative side, preventing degeneracy and reducing redundancy. This adaptive mechanism ensures scalability without sacrificing geometric tightness.

## 5.4 Inference and downstream usage

The overall solving algorithm is summarized in Algorithm 1, and its complexity is discussed in Appendix F. Although the surrogate objective is nonconvex, the update rules in Algorithm 1 exhibit a structured alternating behavior: the weight update is designed to increase the separation gap, while the subsequent gradient step on hyperplane parameters decreases the surrogate objective. This yields a bounded alternating process in which both updates consistently improve the margin-based geometric objective.

Once trained, the convex hull serves as a polyhedral trust region that can be directly deployed in downstream tasks. In **selective classification**, predictions are accepted only if the input lies inside the hull, thereby suppressing overconfident errors outside the data-supported region. In **constraint learning**, the hull provides an explicit linear-feasibility set ($Ax \geq b$) that can be embedded into optimization models, and helps the classifier trained on in-hull samples to capture finer constraint boundaries. This combination of convex filtering and learned approximation improves robustness and reduces complexity in safety-critical applications.

# 6    EXPERIMENTS

## 6.1    DATASET CONSTRUCTION

We evaluate trust-region learning under controlled synthetic geometries and on real-world tabular data. **Synthetic.** We consider three families: (i) PH (polyhedral): we sample $S$ random halfspaces whose intersection is bounded and label points inside as positive; outside are negative. This yields a target hull with at most $S$ facets. (ii) NLC (nonlinear convex): positives form a hypersphere of radius $r$ (convex but non-polyhedral), inducing linear-facet approximation error. (iii) NLNC (nonlinear nonconvex): positives are the union of multiple convex components, testing the ability to tightly envelope nonconvex regions under a facet budget. We denote datasets by type, facet count (if applicable), and dimension, e.g., `PH_15_8`, `NLC_N_10`, `NLNC_N_2`. **Real-world.** We further include representative tabular datasets (e.g., Breast Cancer, Spambase, Bace, HIV).

## 6.2    EXPERIMENTAL SETUP

We compare PCH against: **QH** (Barber et al., 1996), **CGHC** (Barbato et al., 2024), **DCH** (Blum et al., 2019), **PinCHD** (Leng et al., 2019), and **DeepHull** (Balestriero et al., 2022). QH provides an exact geometric reference. CGHC controls hyperplane structure via integer variables. DCH selects sparse representative vertices without using labels, approximating QH but without guaranteeing full positive inclusion. PinCHD serves as a standalone projection-based baseline. DeepHull learns implicit convex regions through neural networks, which do not yield explicit linear constraints ($Ax \geq b$) and do not enforce inclusion of all positives. Accuracy is evaluated in two contexts: *trust region learning* and *downstream classification*. For trust region learning, accuracy is defined as the product of the classification accuracy and the positive inclusion rate. For downstream classification, accuracy refers to the standard test classification accuracy. All experiments are conducted on a server equipped with an Intel(R) Core(TM) i9-10980XE CPU (18 cores, 36 threads) and 64 GB of RAM. Details of the fairness protocol and hyperparameter settings are provided in the Appendix G.

## 6.3    PERFORMANCE ON TRUST REGION LEARNING

We evaluate the performance of different methods to construct accurate and compact trust regions across varying datasets. Table 1 reports the trust region accuracy and model complexity on both synthetic geometries and real-world tabular data. Compared with other methods, PCH consistently achieves high accuracy across all dimensions while maintaining constant model complexity. These results demonstrate that PCH achieves high accuracy, scalability, and interpretability simultaneously. It maintains a tight approximation of the target region while remaining computationally feasible under high-dimensional and large-scale settings. More experimental settings and results are provided in Appendix G.3, G.4, G.5, and G.6.

Further supplementary analyses are provided in the appendix. Appendix G.1 examines the behavior of PCH on linearly separable data, while Appendix G.2 visualizes the full optimization trajectory on the 2D example in Figure 2 and evaluates robustness under different random initializations. Robustness to injected noise in polyhedral datasets is analyzed in Appendix G.3. The effect of the hyperplane budget is illustrated in Figure 15 within Appendix G.4, and Appendix G.5 discusses the behavior of PCH when the positive region is nonlinear and nonconvex. Finally, Appendix G.9 provides additional comparisons with gradient-based baselines as well as a detailed ablation study of key components.

Table 1: Comparison of convex hull learning methods on controlled synthetic geometries and on real-world tabular data. Reported are trust-region accuracy (%) and model complexity (number of hyperplanes). PCH achieves consistent accuracy and controlled complexity, while other methods become inefficient or degrade in high dimensions. A dash "–" indicates that methods fail to scale to the corresponding high-dimensional setting.

| Dimension | Problem | Trust-region Accuracy/% | | | | | | Model Complexity | | | | | |
|---|---|---|---|---|---|---|---|---|---|---|---|---|---|
| | | QH | CGHC | DCH | PinCHD | DeepHull | PCH | QH | CGHC | DCH | PinCHD | DeepHull | PCH |
| Low | PH_15_2 | 100.00 | 99.71 | 97.88 | 99.80 | 97.01 | **100.00** | 37 | 15 | 9 | 11 | 15 | **7** |
| | PH_15_8 | 100.00 | 87.59 | 45.26 | 95.26 | 82.01 | **100.00** | 2.30E+7 | 15 | 6.18E+6 | 480 | 15 | **15** |
| | NLC_N_2 | **100.00** | 99.27 | 95.24 | 99.41 | 97.64 | 99.40 | 93 | 15 | 16 | 24 | 15 | **15** |
| | NLNC_N_2 | **92.15** | 91.83 | 91.87 | 91.86 | 87.05 | 92.10 | 27 | 15 | 27 | 14 | 15 | **15** |
| High | PH_15_100 | – | 57.20 | – | 57.95 | 64.39 | **99.95** | – | 15 | – | 1499 | 15 | **15** |
| | NLC_N_100 | – | 68.89 | – | 53.75 | 75.39 | **99.01** | – | 150 | – | 1499 | 150 | **150** |
| Real-world | Breast Cancer | – | 99.82 | – | 99.82 | 99.82 | **99.82** | – | 5 | – | 10 | 5 | **5** |
| | Spambase | – | 86.70 | – | 91.41 | 91.55 | **94.41** | – | 10 | – | 121 | 11 | **10** |
| | Bace | – | 98.35 | – | 76.80 | 97.37 | **98.61** | – | 5 | – | 149 | 5 | **5** |
| | HIV | – | – | – | – | 99.55 | **99.94** | – | – | – | – | 15 | **15** |

## 6.4 DOWNSTREAM USAGE

We evaluate the benefits of the learned regions in downstream tasks, considering two representative scenarios: selective classification and constraint learning. PCH is compared with CGHC, which also enforces positive coverage while explicitly controlling the hyperplane budget.

**Selective classification.** We combine trust region modeling with base classifiers (CART and MLP) to filter predictions based on the learned convex hulls. A superscript [†] indicates that the classifier is trained on the full dataset and used with trust region filtering during inference; [‡] indicates retraining the classifier using only the samples within the learned trust region. Table 2 reports classification accuracy across five datasets. In most cases, combining classifiers with PCH improves performance compared to standalone classifiers and CGHC variants, confirming its effectiveness in suppressing unreliable predictions outside the trust region. For MLP models retrained under [‡], we apply a reweighting scheme during training to assign higher weights to in-region samples, mitigating potential bias. More experimental settings and classification score results are provided in Appendix G.7.

**Constraint learning.** We evaluate the utility of convex hulls in constraint learning, where the hull serves as a safe approximation of the feasible trust region. Table 3 summarizes feasibility rates and the number of binary variables introduced by different constraint learners. PCH consistently improves reliability while reducing constraint complexity. For example, MLP with PCH[†] increases feasibility from 88% to 94% and reduces the number of binary variables from 205 to 35. The trust region modeling results are compared in Appendix G.5. Detailed grid search results for constraint learning are provided in Appendix G.8. Overall, PCH offers a scalable and reliable trust region for embedding learned constraints into optimization, improving feasibility, simplicity, and computational tractability.

Table 2: Selective classification performance using classifiers trained with or without trust region filtering. [†]: integrated classifier trained on the full dataset. [‡]: integrated classifier retrained based on the learned trust region. PCH consistently improves accuracy across most classifiers and datasets.

| Problem | Trust Region | | CART-based Classification | | | | | MLP-based Classification | | | | |
|---|---|---|---|---|---|---|---|---|---|---|---|---|
| | CGHC | PCH | - | CGHC[†] | CGHC[‡] | PCH[†] | PCH[‡] | - | CGHC[†] | CGHC[‡] | PCH[†] | PCH[‡] |
| NLNC_N_2 | 91.40 | 91.75 | 98.30 | 98.40 | 98.90 | 98.35 | **99.00** | 94.50 | 94.80 | 97.70 | 95.20 | 98.30 |
| breast_cancer | 95.61 | **97.37** | 91.23 | 92.11 | 96.49 | 91.23 | 95.61 | 94.74 | 92.11 | 91.23 | 94.74 | 93.86 |
| spambase | 81.43 | 89.36 | 91.10 | 92.07 | 91.10 | 93.27 | 91.86 | 93.16 | 92.62 | 90.88 | 93.59 | 89.79 |
| bace_ecfp | 70.63 | 78.22 | 74.92 | 70.96 | 72.28 | 75.58 | 77.56 | 74.26 | 72.28 | 72.94 | 77.89 | **78.55** |
| hiv_ecfp | – | 95.68 | 95.05 | – | – | **96.89** | 95.81 | 95.76 | – | – | 96.77 | 96.13 |

Table 3: Constraint learning and embedding performance using models trained with or without trust region filtering. [†]: integrated constraint learning model trained on the full dataset. [‡]: integrated model retrained based on the learned trust region. PCH improves both feasibility and model simplicity.

| Index | No rule | CART-based constraint learning | | | | | MLP-based constraint learning | | | | |
|---|---|---|---|---|---|---|---|---|---|---|---|
| | | - | CGHC[†] | CGHC[‡] | PCH[†] | PCH[‡] | - | CGHC[†] | CGHC[‡] | PCH[†] | PCH[‡] |
| Feasibility Rate(%) | 49.00 | 69.00 | 70.00 | 57.00 | 84.00 | 72.00 | 88.00 | 88.00 | 90.00 | **94.00** | 93.00 |
| Binary Variable Number | 0 | 12 | 33 | 50 | **12** | 28 | 205 | 55 | 35 | 35 | 35 |

## 7 CONCLUSION

This paper introduces Projection Convex Hull (PCH), a principled framework for trust region learning that connects the exact MINLP formulation with an unconstrained surrogate objective. Through subregion decomposition and weight assignment, PCH provides theoretical guarantees, while its algorithm, combining subregion partition, weight projection, and gradient-based updates, yields scalable, adaptive, and boundary-tight convex hulls.

Experiments on synthetic and real-world datasets show that PCH achieves consistent tightness and scalability, while remaining compact and interpretable. Beyond standalone learning, the learned hulls benefit downstream tasks such as selective classification and constraint learning, confirming the practical value of theoretically grounded trust region modeling. Future work may extend PCH to nonconvex decompositions for broader safety-critical applications.

## ETHICS STATEMENT

This paper focuses on theoretical modeling and algorithmic development for convex hull–based trust region learning. All experiments are conducted on publicly available benchmark datasets (e.g. Breast Cancer, Spambase, Bace, and HIV), which do not contain private or sensitive information. The proposed methodology does not raise concerns regarding human subjects, discrimination, security, or other ethical issues. We have carefully adhered to the ICLR Code of Ethics in conducting and reporting this research.

## REPRODUCIBILITY STATEMENT

The theoretical foundation of Projection Convex Hull (PCH) is fully developed in Section 4, with all proofs provided in Appendix C. The complete algorithmic procedure is described in Section 5, with detailed pseudocode in Algorithm 1. Experimental settings, datasets, and hyperparameter configurations are reported in Appendix G. All benchmark datasets used are publicly available, and the construction of synthetic datasets follows explicitly specified procedures. Together, these elements provide a comprehensive and transparent description that allows independent reproduction of our results.

### ACKNOWLEDGMENTS

This work was supported by the National Key R&D Program of China under grant 2025YFE0110900.

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

APPENDIX / SUPPLEMENTAL MATERIAL

## A    NOTATIONS

In the main text and the appendix, we use the following notations:

Table 4: Summary of notations used in the paper.

| Symbol | Description |
|---|---|
| $\mathbb{R}^d$ | $d$-dimensional real space |
| $\|a\|$ | Euclidean norm of vector $a$, i.e., $\sqrt{\sum_j a_j^2}$ |
| $\text{conv}(\cdot)$ | Convex hull of a set of points |
| $\text{partition}_{(d)}(\cdot)$ | $d$-th order statistic (the $d$-th smallest value) of the input |
| $\mathcal{D}$ | Labeled dataset $\{(x_i, y_i)\}_{i=1}^n$, $x_i \in \mathbb{R}^d$, $y_i \in \{-1, +1\}$ |
| $\mathcal{D}_+, \mathcal{D}_-$ | Positive and negative subsets of $\mathcal{D}$ |
| $\mathcal{D}'_-$ | Negatives lying outside $\text{conv}(\mathcal{D}_+)$ |
| $\mathcal{C}$ | Learned polyhedral convex region (trust region) |
| $S$ | Hyperplane budget (maximum number of facets) |
| $(a^s, b^s)$ | Normal vector and bias of the $s$-th hyperplane |
| $b^\star(a)$ | Optimal offset for $a$, defined as $b^\star(a) = \min_{x \in \mathcal{D}_+} a^\top x$ |
| $A = [a^1, \dots, a^S]$ | Matrix of hyperplane normals |
| $b = [b^1, \dots, b^S]$ | Vector of hyperplane offsets |
| $A^{-s}, b^{-s}$ | Hyperplane normals/offsets excluding the $s$-th facet |
| $\mathcal{D}_-^s$ | Subset of negatives assigned to hyperplane $s$ |
| $\mathcal{D}^s$ | Local subregion used for optimizing hyperplane $s$ |
| $\underline{m}^s, \overline{m}^s$ | Lower/upper thresholds for subregion band selection |
| $m_{\textbf{shift}}$ | Hyperparameter controlling bias of $\overline{m}^s$ |
| $\xi^s$ | Separation margin of hyperplane $s$ |
| $\xi_i^s$ | Slack variable for negative sample $x_i$ w.r.t. hyperplane $s$ |
| $z_i^s$ | Binary variable indicating whether negative sample $x_i$ is separated by hyperplane $s$. |
| $f(a, b)$ | Surrogate objective for hyperplane $(a, b)$ |
| $p_i^{\textbf{L}}, p_i^{\textbf{R}}$ | Left/right soft split probabilities for sample $x_i$ |
| $\beta$ | Temperature parameter controlling softness of split |
| $\omega_i$ | Weight assigned to sample $x_i$ |
| $\omega$ | Vector of sample weights $[\omega_i]$ |
| $C\omega = d$ | Linear constraints on $\omega$ ensuring surrogate stationarity |
| $K_{\pi+}, K_{\pi-}$ | Isotonic cones enforcing monotonic weights within positives/negatives |
| $P_S, P_+, P_{K_{\pi\pm}}$ | Projection operators onto linear, nonnegative, and isotonic constraints |
| $x_+, x_-$ | Boundary-supporting positive and negative samples in subregion $\mathcal{D}^s$ |

## B    MILP FORMULATION OF PCAB

To formalize the task of learning a polyhedral convex region that encloses all positives while excluding as many negatives as possible, we recall the *Polyhedral Convex Hull Approximation with Budget* (PCAB) formulation introduced in Barbato et al. (2024).

Given a dataset $\mathcal{D} = \mathcal{D}_+ \cup \mathcal{D}_-$, the goal is to construct a convex region

$$\mathcal{C} = \{\, x \in \mathbb{R}^d \ \mid \ a^{s\top} x \geq b^s, \ s = 1, \dots, S \,\}$$

with at most $S$ hyperplanes, such that $\mathcal{D}_+ \subseteq \mathcal{C}$ and the number of negatives from $\mathcal{D}_-$ incorrectly included in $\mathcal{C}$ is minimized. This leads to the following MILP:

$$
\begin{aligned}
\min \quad & \sum_{x_i \in \mathcal{D}_-} z_i \\
\text{s.t.} \quad & \sum_{s=1}^{S} z_i^s = 1 - z_i, && \forall\, x_i \in \mathcal{D}_-, \\
& a^{s\top} x_i - b^s \geq 1, && \forall\, x_i \in \mathcal{D}_+, \ \forall s, \\
& a^{s\top} x_i - b^s \leq -1 + \xi_i^s, && \forall\, x_i \in \mathcal{D}_-, \ \forall s, \\
& \xi_i^s \leq M \cdot (1 - z_i^s), && \forall\, x_i \in \mathcal{D}_-, \ \forall s, \\
& \xi_i^s \geq 0, && \forall\, x_i \in \mathcal{D}_-, \ \forall s, \\
& z_i^s \in \{0, 1\}, && \forall\, x_i \in \mathcal{D}_-, \ \forall s, \\
& z_i \in \{0, 1\}, && \forall\, x_i \in \mathcal{D}_-.
\end{aligned}
\tag{12}
$$

where $z_i$ is a binary indicator of whether negative sample $x_i$ is included in the learned hull $\mathcal{C}$. $z_i^s$ is an auxiliary binary variable indicating whether $x_i$ is separated by hyperplane $s$. $\xi_i^s$ is a nonnegative slack variable capturing violations of separation for $x_i$ w.r.t. hyperplane $s$. The big-$M$ constraint ensures that slack $\xi_i^s$ is only active when $x_i$ is not separated by hyperplane $s$.

All positives are required to satisfy $a^{s\top} x_i - b^s \geq 1$ for every facet $s$, thereby guaranteeing their enclosure within $\mathcal{C}$.

**Discussion of properties.** Formulation Equation 12 allows explicit modeling of negative inclusion and is compatible with generic MILP solvers (e.g., Gurobi). However, it does not inherently control the *geometric tightness* of the hull: 1) The constraints impose a fixed separation band of width 1 between positives and negatives. Hyperplanes may therefore be positioned conservatively, leaving unused gaps between $\mathcal{C}$ and $\mathrm{conv}(\mathcal{D}_+)$. 2) The optimization objective depends only on the binary inclusion of negatives ($z_i$), but not on how closely facets align with the positive class boundary. 3) Multiple optimal solutions typically exist with the same objective but very different geometric tightness. This non-uniqueness arises because scaling or shifting hyperplanes can preserve feasibility and the binary objective value, yet yield looser or tighter hulls.

**Proposition 1.** *The MILP-based PCAB formulation admits infinitely many optimal solutions in linearly or polyhedrally separable cases, with potentially large variation in boundary tightness.*

*Proof.* In a linearly separable case, suppose a hyperplane $(a^\star, b^\star)$ perfectly separates positives and negatives with zero slack. Then the MILP objective is zero. However, any scaled version $(\alpha a^\star, \alpha b^\star + \epsilon)$ with $\alpha > 1$ and $\epsilon \in [1 - \alpha, \alpha - 1]$ also satisfies all constraints and yields the same objective, producing infinitely many optimal but geometrically distinct solutions.

In a polyhedrally separable case with $S$ hyperplanes, let $\{(a^{s\star}, b^{s\star})\}_{s=1}^S$ define an optimal separating hull. Replacing $(a^{s\star}, b^{s\star})$ with $(2a^{s\star}, 2b^{s\star} + 1)$ maintains feasibility (same separation margins) and preserves the objective, again yielding different feasible optima with varying tightness. $\qquad\square$

**Implication for trust region learning.** Although MILP formulations are powerful for minimizing negative inclusion, they are not sufficient for applications where the trust region must be *boundary-tight*. In safety-critical or generalization-sensitive tasks, the hull should closely align with $\mathrm{conv}(\mathcal{D}_+)$, not merely avoid negatives. The rigid band constraints in Equation 12 prevent this alignment, especially when $S$ is small or the true boundary is nonlinear. Additionally, their computational complexity grows quickly with both sample size and dimension: each additional negative sample introduces binary variables $z_i^s$, and the resulting search space is exponential in $|\mathcal{D}_-| \times S$. In practice, directly solving Equation 12 becomes intractable even for small-scale datasets.

This motivates our surrogate-based approach: rather than enforcing separation through hard binary variables, we design a differentiable loss that directly guides hyperplanes toward the positive boundary, enabling compact, adaptive, and geometrically tight trust regions.

# C PROOFS FOR THEORETICAL FOUNDATION

## C.1 PROOF OF LEMMA 1.

Consider the global MINLP formulation in Equation 2, which minimizes the total margin penalties $\sum_s \xi^s$ under coverage and separation constraints. Let $(A^\star, b^\star)$ denote its optimal solution. By construction, each negative sample $x_i \in \mathcal{D}_-$ must be assigned to at least one hyperplane, i.e., $z_i^s = 1$. Define $\mathcal{D}_-^s = \{x_i \in \mathcal{D}_- : z_i^s = 1\}$ as the set of negatives excluded by hyperplane $s$.

Now fix a particular hyperplane $s$. The pair $(a^{s\star}, b^{s\star})$ together with $\xi^{s\star}$ must satisfy all constraints in the local separation problem Equation 3 restricted to $\mathcal{D}_+ \cup \mathcal{D}_-^s$. Suppose, for contradiction, that there exists another feasible $(a', b')$ with strictly larger separation margin $\xi' > \xi^{s\star}$. Replacing $(a^{s\star}, b^{s\star})$ by $(a', b')$ in the global solution would reduce the overall objective of Equation 2, contradicting its optimality. Hence $(a^{s\star}, b^{s\star})$ must also be optimal for the corresponding subproblem Equation 3. $\quad\square$

## C.2   Proof of Lemma 2.

Fix a local dataset $\mathcal{D}^s = \mathcal{D}_+ \cup \mathcal{D}_-^s$ and consider the surrogate Equation 4. Define

$$A := \sum_i \omega_i p_i^{\mathrm{R}}, \quad B := \sum_i \omega_i p_i^{\mathrm{R}} y_i, \quad N := \sum_i \omega_i, \quad E := \sum_i \omega_i y_i.$$

Using $p_i^{\mathrm{L}} = 1 - p_i^{\mathrm{R}}$, the surrogate can be rewritten as

$$f(a,b) = -\tfrac{1}{2}\left( \frac{B^2}{A} + \frac{(E-B)^2}{N-A} \right). \tag{13}$$

**Step 1: Stationarity in the $b$-direction.** We have $\partial_b p_i^{\mathrm{R}} = -(\beta/\|a\|)\, p_i^{\mathrm{R}} p_i^{\mathrm{L}}$, hence

$$\partial_b A = -\tfrac{\beta}{\|a\|} \sum_i \omega_i p_i^{\mathrm{R}} p_i^{\mathrm{L}}, \qquad \partial_b B = -\tfrac{\beta}{\|a\|} \sum_i \omega_i y_i p_i^{\mathrm{R}} p_i^{\mathrm{L}}.$$

By the chain rule applied to Equation 13,

$$\partial_b f = -\tfrac{1}{2}\left( \frac{2B(\partial_b B)\,A - B^2(\partial_b A)}{A^2} + \frac{2(E-B)(-\partial_b B)\,(N-A) - (E-B)^2(-\partial_b A)}{(N-A)^2} \right).$$

At $(a^\star, b^\star)$, choose weights such that

$$E = 0, \quad N = 2, \quad A = 1, \quad \sum_i \omega_i y_i p_i^{\mathrm{R}} p_i^{\mathrm{L}} = 0. \tag{14}$$

Then $\partial_b B = 0$ and $A = N - A = 1$, so the two fractions cancel and $\partial_b f(a^\star, b^\star) = 0$. Since $b^\star$ is placed at the positive support $b_{a^\star}^\star := \min_{x \in \mathcal{D}_+} a^{\star\top} x$, these linear constraints can enforce boundary-tight stationarity along $b$ at $(a^\star, b^\star)$.

**Step 2: Stationarity in the $a$-direction.** Simplifying the gradient of Equation 13 under $E = 0, N = 2, A = 1$ yields

$$\partial_a f = -\frac{2\beta}{\|a\|} B \underbrace{\sum_i \omega_i\, p_i^{\mathrm{L}} p_i^{\mathrm{R}}\, y_i\left(x_i - \tfrac{a^\top x_i}{\|a\|^2}a\right)}_{=:\, C'\omega}. \tag{15}$$

Hence $\partial_a f = 0$ whenever $C'\omega = 0$, where $C' = [p_i^{\mathrm{L}} p_i^{\mathrm{R}} y_i\left(x_i - \tfrac{a^\top x_i}{\|a\|^2}a\right)]_{x_i \in \mathcal{D}^s}$.

**Step 3: Existence of weights.** If the weight vector $\omega$ satisfies the linear constraints $C\omega = d$, then by Step 1 we have $\partial_b f(a^\star, b^\star) = 0$. By Step 2, $\partial_a f(a^\star, b^\star) = 0$ holds whenever $C'\omega = 0$.

To see that such $\omega$ exists, stack the constraints as a single linear system

$$C^*\omega = d^*, \qquad C^* := \begin{bmatrix} C \\ C' \end{bmatrix}, \qquad d^* := \begin{bmatrix} d \\ 0 \end{bmatrix},$$

where $C \in \mathbb{R}^{4 \times n}$ and $d \in \mathbb{R}^4$ encode the four scalar equalities in Equation 5 or Equation 14, and $C' \in \mathbb{R}^{d \times n}$ has columns $p_i^{\mathrm{L}} p_i^{\mathrm{R}} y_i\left(x_i - \tfrac{a^{\star\top} x_i}{\|a^\star\|^2}a^\star\right)$. If $\mathcal{D}^s$ contains at least $d+4$ near-boundary points from each class in general position, the row space of $C^* \in \mathbb{R}^{(d+4) \times n}$ is generically full, and a solution $\omega$ to $C^*\omega = d^*$ exists (not necessarily unique). This solution simultaneously satisfies $C\omega = d$ (so $\partial_b f = 0$) and $C'\omega = 0$ (so $\partial_a f = 0$), making $(a^\star, b^\star)$ a stationary point of the surrogate.

*Constructive sketch.* Suppose we can freely choose samples from the half-space on both sides of the optimal hyperplane $a^{\star\top} x = b_a^\star$. We can get $x_{i+}$ and $x_{j-}$ such that they are symmetric about the optimal hyperplane, i.e., there exists $m > 0$ such that

$$\frac{a^{\star\top} x_{i+} - b_a^\star}{\|a^\star\|} = m, \qquad \frac{a^{\star\top} x_{j-} - b_a^\star}{\|a^\star\|} = -m.$$

Assign weights $\omega_{i+} = \omega_{j-} = 1$ and $\omega_i = 0$ for all other $i$. Then

$$\sum \omega_i = 2, \quad \sum \omega_i y_i = 0, \quad \sum \omega_i p_i^{\mathrm{R}} = 1, \quad \sum \omega_i p_i^{\mathrm{L}} p_i^{\mathrm{R}} y_i = 0,$$

Therefore $C\omega = d$ holds.

Since $x_{i+}$ and $x_{j-}$ are symmetric about the optimal hyperplane, i.e., $x_{i+} - x_{j-} = k\, a^\star$ for $k \in \mathbb{R}$. Consequently,

$$C'\omega = p_{i+}^{\mathrm{L}} p_{i+}^{\mathrm{R}} \left( x_{i+} - x_{j-} - \frac{a^{\star\top}(x_{i+} - x_{j-})}{\|a^\star\|^2} a^\star \right) = p_{i+}^{\mathrm{L}} p_{i+}^{\mathrm{R}} \left( k\, a^\star - \frac{a^{\star\top} k\, a^\star}{\|a^\star\|^2} a^\star \right) = 0,$$

which establishes $C'\omega = 0$. Combining $C\omega = d$ and $C'\omega = 0$ yields both $b$- and $a$-direction stationarity at $(a^\star, b^\star)$. □

### C.3  PROOF OF LEMMA 3.

The linear constraints $C\omega = d$ ensure that $\partial_b f(a, b) = 0$ and the gradient of Equation 4 can be deduced into Equation 15, which can be re-written as:

$$\partial_a f(a, b) = -\frac{2\beta}{\|a\|} B \left( \sum_i \omega_i\, p_i^{\mathrm{L}} p_i^{\mathrm{R}}\, y_i\, x_i - \frac{a^\top \left( \sum_i \omega_i\, p_i^{\mathrm{L}} p_i^{\mathrm{R}}\, y_i\, x_i \right)}{\|a\|^2} a \right). \tag{16}$$

The term inside the parentheses corresponds exactly to the fixed-point mapping in Equation 6, namely

$$a = \sum \omega_i\, p_i^{\mathrm{L}} p_i^{\mathrm{R}}\, y_i\, x_i.$$

If $(a, b)$ and $\{\omega_i\}$ satisfy Equation 6, then the first term vanishes up to a projection along $a$, leaving

$$\partial_a f(a, b) \;\propto\; a - \frac{a^\top a}{\|a\|^2} a \;=\; 0.$$

Therefore, under the joint conditions of the fixed-point relation Equation 6 and $C\omega = d$, both $\partial_a f = 0$ and $\partial_b f = 0$ hold, so $(a, b)$ is a stationary point of the surrogate objective. □

### C.4  PROOF OF THEOREM 1.

Let $(A^\star, b^\star)$ be an optimal solution of the MINLP Equation 2. By Lemma 1, for each facet $s \in \{1, \dots, S\}$ the pair $(a^{s\star}, b^{s\star})$ solves the local constrained problem Equation 3 on $\mathcal{D}^s = \mathcal{D}_+ \cup \mathcal{D}_-^s$.

Fix any $s$. By Lemma 2, there exists a weight assignment on $\mathcal{D}^s$ such that $(a^{s\star}, b^{s\star})$ is a stationary point of the surrogate objective Equation 4. As shown in the proof of Lemma 2 (Step 3), such weights can be obtained by solving a linear system that stacks the $b$-direction constraints Equation 5 and the $a$-direction normal-equation constraints, and a feasible solution exists under near-boundary coverage assumptions on $\mathcal{D}^s$. Moreover, Lemma 3 provides a sufficient condition for the $a$-direction stationarity when the optimal $a^{s\star}$ is unknown: if the fixed-point relation Equation 6 holds together with Equation 5, then $(a^{s\star}, b^{s\star})$ is stationary for Equation 4.

Therefore, for each hyperplane $s$ there exists a (possibly non-unique) weight assignment $\{\omega_i^s\}$ making $(a^{s\star}, b^{s\star})$ a stationary point of the surrogate objective. This proves the claim. □

## D  FEASIBILITY OF THE PROPOSED METHODOLOGY

The surrogate formulation in Section 4 reduces the MINLP into a weight assignment problem. In principle, one could solve this problem by repeatedly optimizing $f(a, b)$ until convergence, enforcing $a \approx g(a, \omega)$, and then updating $\omega$ so as to maximize the margin $\xi(\omega)$. During the process, one should immediately conduct projection through $P_S(\omega)$ to ensure $C\omega = d$ after each updating of $a$ and $\omega$. This would give an iterative scheme directly aligned with the constrained hyperplane problem Equation 3. However, such an approach is computationally expensive and numerically unstable: every update of $\omega$ requires re-solving $f(a, b)$ to convergence, and small perturbations in $\omega$ may lead to oscillatory or divergent behavior.

**Need for additional constraints.** To make weight updates meaningful, we introduce further constraints beyond $C\omega = d$: **1) Nonnegativity** ($\omega \geq 0$) ensures that weights behave as proportions of contribution; negative weights have no physical interpretation; **2) Isotonicity** ($\omega \in K_{\pi+}, K_{\pi-}$) enforces that samples closer to the boundary receive larger weights, consistent with the role of support vectors. These constraints encode the geometric intuition of the problem and regularize the update process, preventing $\omega$ from drifting into unrealistic assignments that would destabilize $f(a, b)$.

**Projection onto multiple constraints.** Once these constraints are included, a single projection is insufficient, because the feasible set is the intersection of four convex sets ($C\omega = d$, $\omega \geq 0$, $K_{\pi+}$, $K_{\pi-}$). We therefore rely on alternating projections (Dykstra's algorithm) to enforce feasibility after each gradient update. This guarantees that the weight assignments remain consistent with both theoretical requirements and geometric intuition.

**Approximation for efficiency.** A fully exact implementation would require: (i) letting $f(a, b)$ converge at each step, and (ii) applying Dykstra's algorithm until its own convergence. In practice this is prohibitively costly. Instead, we adopt a pragmatic approximation: perform only one gradient step on $f(a, b)$, one gradient step on $\xi(\omega)$, and then one round of projections. Although this does not ensure exact satisfaction of the fixed-point relation $a = g(a, \omega)$, empirical evidence shows that the resulting $a$ remains close to the fixed point, while the projections ensure $\omega$ stays feasible and interpretable. This strikes a balance between theoretical rigor and computational tractability.

**Summary.** While the approximation $\partial a / \partial \omega \approx \partial g / \partial \omega$ is not exact, empirical results show that the resulting updates produce stable weight distributions and lead to boundary-tight hyperplanes. Moreover, the incremental update scheme (one step on $f(a, b)$ followed by one projection of $\omega$) is computationally efficient, avoiding repeated full convergence of $f(a, b)$ at every iteration. This trade-off between exact fixed-point satisfaction and practical efficiency makes the methodology feasible for large-scale, high-dimensional problems.

# E  IMPROVED HYPERPLANE INITIALIZATION VIA RELIABLE NEGATIVE SAMPLE SELECTION

The quality of hyperplane initialization directly affects the convergence and stability of convex hull learning algorithms. Algorithm 2 describes the improved initialization strategy used in our framework.

This strategy is adapted from the PinCHD method originally proposed in Leng et al. (2019), which constructs supporting hyperplanes by iteratively computing the closest point between a negative sample and the convex hull of positive samples. The resulting support vector pair is then used to form an initial hyperplane. This enables fast initialization of polyhedral trust regions. However, the original method is sensitive to the choice of the negative sample: if the selected negative lies inside $\text{conv}(\mathcal{D}_+)$, the projection becomes ill-posed, producing degenerate hyperplanes or leading to singularities during subsequent optimization.

To address this, we propose a more robust initialization strategy that carefully restricts the choice of negative anchor samples. Specifically, the selected sample must satisfy two conditions: 1) it lies inside the current polyhedral region

$$\mathcal{C} = \{x \in \mathbb{R}^d \mid a^{s\top} x \geq b^s, \ \forall s\},$$

so that it corresponds to a misclassified negative that ideally should have been excluded, guiding the algorithm to focus on problematic regions; 2) it belongs to at least one local subregion $\mathcal{D}^s$ defined in Section 4, ensuring that the anchor is close to the decision boundary and more likely to serve as a meaningful support vector.

---

**Algorithm 2** Trustworthy Hyperplane Initialization (Improved PinCHD)

---

**Require:** Positive set $\mathcal{D}_+$, local partitions $\{\mathcal{D}^s\}_{s=1}^S$, current convex hull $\mathcal{C}$, tolerance $\epsilon$
**Ensure:** A supporting hyperplane $(a^s, b^s)$

1: Randomly select a negative sample $x^- \in \left(\bigcup_{s=1}^S \mathcal{D}_-^s\right) \cap \mathcal{C}$
2: Initialize $x_{\text{new}}^* \in \mathcal{D}_+$
3: **repeat**
4:     $x^* \leftarrow x_{\text{new}}^*$
5:     $x_t \leftarrow \arg\min_{x \in \mathcal{D}_+} \frac{(x - x^-)^\top (x^* - x^-)}{\|x^* - x^-\|}$
6:     $q \leftarrow \min\left(1, \frac{(x^* - x^-)^\top (x^* - x_t)}{\|x^* - x_t\|^2}\right)$
7:     $x_{\text{new}}^* \leftarrow x^* + q(x_t - x^*)$
8: **until** $\|x^* - x^-\| - \|x_{\text{new}}^* - x^-\| < \epsilon$
9: $a \leftarrow (x_{\text{new}}^* - x^-)/\|x_{\text{new}}^* - x^-\|$
10: $b \leftarrow \min_{x \in \mathcal{D}_+}(a^\top x)$
11: **if** $\exists x \in \mathcal{D}_+$ such that $a^\top x - b < -\epsilon$ **then**
12:     Randomly re-initialize $a$ with $\|a\| = 1$, set $b = \min_{x \in \mathcal{D}_+}(a^\top x)$
13: **end if**
14: **return** $(a, b)$

---

Once the negative sample $x^-$ is selected, the algorithm applies an iterative projection process to find the closest point $x_{\text{new}}^* \in \text{conv}(\mathcal{D}_+)$ to $x^-$. This is achieved by repeatedly projecting $x^-$ onto line segments between $x^*$ and candidate positives $x_t$, until convergence within a tolerance $\epsilon$. The resulting vector $a = x_{\text{new}}^* - x^-$ defines the normal of the hyperplane, normalized to unit length. The bias $b$ is then set as $b = \min_{x \in \mathcal{D}_+}(a^\top x)$, ensuring tangency to the positive hull and that all positives remain inside $\mathcal{C}$.

To safeguard against numerical instability, we check whether all positives satisfy $a^\top x - b \geq -\epsilon$. If this fails (which typically indicates that $x^-$ lies within $\text{conv}(\mathcal{D}_+)$), we fallback to a random initialization. This ensures robustness against degenerate anchors.

In summary, our improved initialization strategy selects only *reliable* negatives and guarantees geometrically meaningful hyperplanes. This avoids degenerate configurations, stabilizes training, and provides a strong starting point for the surrogate-based optimization in the main framework.

## F  COMPLEXITY ANALYSIS OF THE PROPOSED ALGORITHM

We analyze the computational complexity of our PCH framework, as described in Algorithm 1. Let $n = |\mathcal{D}|$ denote the dataset size, $d$ the input dimension, and $S$ the maximum number of hyperplanes. We examine each component in turn.

**1) Initialization via improved PinCHD (Algorithm 2).** The improved PinCHD algorithm initializes each hyperplane by projecting a selected negative sample $x^- \in \mathcal{D}_-^s$ toward $\text{conv}(\mathcal{D}_+)$. In each iteration, all positive samples are scanned to identify the closest supporting point, which requires $\mathcal{O}(n_+ d)$ operations. Assuming convergence within $T$ iterations (empirically a small constant), the cost per hyperplane is $\mathcal{O}(Tn_+ d)$. Computing the bias $b^s = \min_{x \in \mathcal{D}_+} a^{s\top} x$ requires $\mathcal{O}(n_+ d)$ additional operations. Thus, the total initialization cost across all $S$ hyperplanes is $\mathcal{O}(TSn_+ d)$.

**2) Bias computation.** At each iteration, the offset $b_{a^s}^\star$ is recomputed for every hyperplane, requiring $\min_{x \in \mathcal{D}_+} a^{s\top} x$. The per-hyperplane cost is $\mathcal{O}(n_+ d)$, yielding $\mathcal{O}(Sn_+ d)$ in total.

**3) Subregion assignment (Equation 7).** For each hyperplane $s$, we must identify the local dataset $\mathcal{D}^s$ by checking whether $a^{s\top} x - b_{a^s}^\star$ lies within the band $[-\overline{m}^s, \overline{m}^s]$ and whether $x$ satisfies the other $S - 1$ hyperplane constraints. This requires computing the signed margins $\{a^{s\top} x - b^s\}_{s=1}^S$ for all $n$ samples, at cost $\mathcal{O}(nSd)$. Computing $\overline{m}^s$ requires evaluating $a^{s\top} x$ across $n_+$ positives and selecting the $d$-th order statistic, i.e., $\mathcal{O}(n_+ d)$ per hyperplane. The overall complexity of subregion assignment is therefore $\mathcal{O}(nSd)$.

**4) Surrogate evaluation and gradient computation (Equation 4).** Within each subregion $\mathcal{D}^s$, evaluating the surrogate $f(a^s, b^s)$ requires computing the soft-split terms $p_i^{\mathrm{L}}, p_i^{\mathrm{R}}$ and aggregating weighted sums. This requires $\mathcal{O}(n_s d)$ operations, where $n_s = |\mathcal{D}^s|$. Across all hyperplanes, the total cost is $\mathcal{O}(d \sum_{s=1}^{S} n_s)$, bounded above by $\mathcal{O}(nSd)$.

**5) Weight update and projections (Equation 9).** The gradient $\nabla_\omega \xi(\omega)$ requires weighted sums of samples in $\mathcal{D}^s$, which is $\mathcal{O}(n_s d)$. After the gradient step, the updated $\omega$ must be projected back onto the feasible set

$$\mathcal{F} = \{\, \omega \geq 0, \ \omega \in K_{\pi^+}, \ \omega \in K_{\pi^-}, \ C\omega = d \,\}.$$

Each projection is efficient: 1) $P_+$ (nonnegativity): $\mathcal{O}(n_s)$, 2) $P_{K_{\pi^+}}, P_{K_{\pi^-}}$ (isotonic projections via PAVA): $\mathcal{O}(n_s)$ each, 3) $P_S$ (linear equalities): $\mathcal{O}(n_s)$, since it reduces to a rank-4 update. Thus the overall projection cost is $\mathcal{O}(n_s)$ per hyperplane, and $\mathcal{O}(\sum_{s=1}^{S} n_s)$ across all hyperplanes. The dominant cost remains in $\nabla_\omega \xi(\omega)$, i.e., $\mathcal{O}(nSd)$.

**6) Gradient descent step for $(a^s, b^s)$.** Updating each hyperplane requires $\mathcal{O}(d)$ operations for $a^s \in \mathbb{R}^d$ and constant time for $b^s$. Thus the total cost across $S$ hyperplanes is $\mathcal{O}(Sd)$, which is negligible compared to surrogate evaluation.

**Overall Complexity.** The total computational cost of PCH consists of a one-time initialization phase plus multiple gradient-based iterations. Initialization requires $\mathcal{O}(TSn_+d)$. Each iteration involves subregion assignment, surrogate evaluation, weight updates, and parameter updates, dominated by $\mathcal{O}(nSd)$. For $I$ iterations, the overall complexity is therefore $\mathcal{O}(InSd)$, which scales linearly in dataset size, feature dimension, number of hyperplanes, and iteration count.

## G  EXPERIMENTAL DETAILS AND ADDITIONAL RESULTS

### G.1  LINEAR SEPARATION

**Dataset generation and training settings.** This experiment evaluates the ability of our proposed method to learn a linear separator between two classes. The dataset is a synthetic 2D dataset containing 19,979 samples, randomly generated such that the positive class (blue) and negative class (red) are linearly separable, as illustrated in Figure 4. We compare our method (PCH) with the baseline `SGDClassifier` from `scikit-learn`, denoted as SGD. Both models are trained for 50 epochs of gradient descent with a learning rate of 0.01, where each epoch computes the full-batch gradient over the entire dataset. This ensures a fair comparison under consistent update schedules and optimization dynamics.

**Visualization of training.** Figure 4 visualizes the learned hyperplanes at different iterations. Our method (left) progressively refines the hyperplane direction while consistently positioning it near the boundary of the positive class, resulting in rapid convergence toward the maximum-margin orientation. In contrast, although the SGD baseline (right) also minimizes a hinge loss and thus seeks a large-margin separator, it jointly updates both orientation and bias without explicit geometric constraints. As a result, its convergence is slower, and the learned hyperplane often remains offset from the true boundary.

**Training dynamics.** Figure 5 reports the training accuracy and signed separation margin over iterations. We define the signed separation margin (or gap) as $\max_{x_i \in \mathcal{D}_-} (a^\top x_i - b)$, which measures the maximum signed distance from negative samples to the hyperplane. Since the learned hyperplane is always positioned to the positive class boundary, this metric effectively captures the tightness of negative exclusion. Our method steadily improves classification accuracy while simultaneously minimizing the gap, indicating that the learned separator is both accurate and geometrically tight. In contrast, SGD quickly achieves high accuracy but converges to a hyperplane with a larger margin violation, reflecting weaker control over the geometric alignment of the boundary.

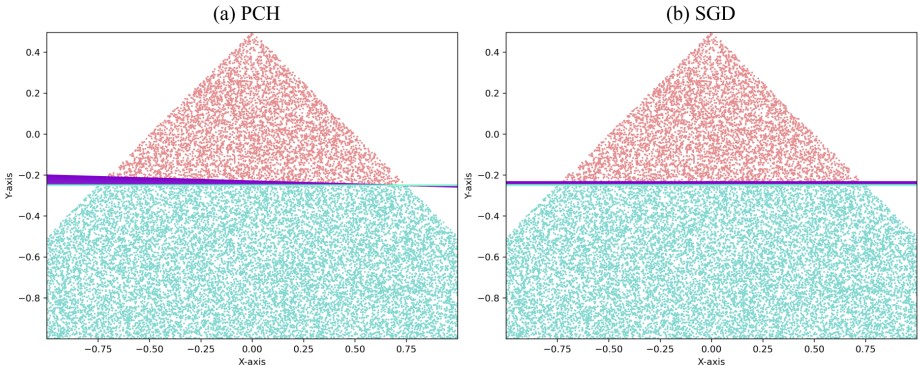

Figure 4: Evolution of the supporting hyperplane during training. Left: our method (PCH); Right: SGD baseline.

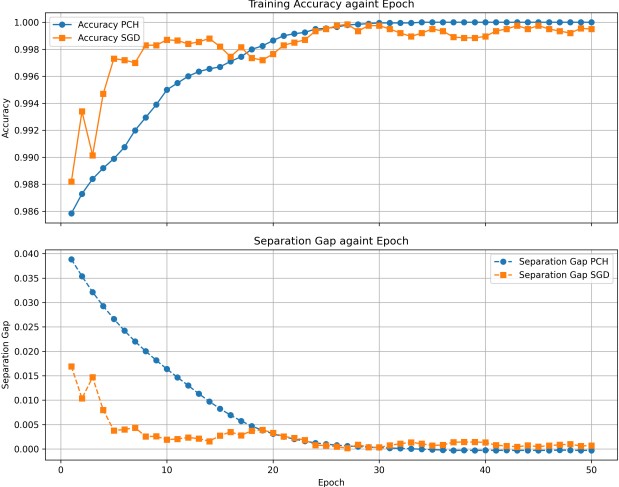

Figure 5: Training accuracy and signed separation gap over training epochs.

**Main results.**  We also report the final results in Table 5, including classification accuracy, runtime, and the signed separation gap. The results confirm that our method (PCH) achieves perfect accuracy while producing a hyperplane that is nearly tangent to the positive class boundary. The negative gap evidences the performance of PCH in excluding negative samples with a bigger separation margin. In contrast, although the SGD baseline reaches comparable accuracy, it converges to a solution with a slightly worse geometric margin, indicating less precise alignment with the optimal separating boundary. The runtimes of both methods are comparable, suggesting that the additional structure imposed by PCH does not introduce significant computational cost.

Table 5: Final comparison on linearly separable task

| Method | Accuracy (%) | Runtime (s) | Signed Separation Gap |
|--------|--------------|-------------|-----------------------|
| PCH    | 100.00       | 0.180       | -0.00030              |
| SGD    | 99.95        | 0.188       | +0.00072              |

**Summary.**  This experiment demonstrates that our proposed method not only matches the classification accuracy of standard SGD-based training on linearly separable data, but also achieves superior geometric alignment. By explicitly fixing the bias term to the boundary of the positive class

and updating the hyperplane orientation based on gradient descent on a projected-weight surrogate loss, PCH offers a more stable and interpretable mechanism for learning individual tight separation hyperplanes, laying a solid foundation for constructing compact convex hulls in more complex settings.

## G.2 TRIANGLE SEPARATION FOR VISUALIZATION

We present the full training trajectory of PCH on the same 2D dataset used in Figure 2. This extended example serves four purposes: (1) visualizing how the surrogate optimization evolves over iterations, (2) empirically demonstrating the convergence of both hyperplane parameters and weight assignments, (3) examining the robustness of the optimization with respect to different initializations, and (4) illustrating how the learned polyhedral separator relates to the MINLP formulation.

**Dataset generation and training settings.** We construct a simple 2D dataset on a uniform grid over the square $[-1, 1]^2$. A triangular positive region is generated by selecting three supporting directions and forming a regular polygonal boundary. All grid points inside this region are labeled as positive, and the remaining points are labeled as negative. This produces a clean binary classification task with a well-defined geometric structure. We train a PCH model with three hyperplanes for 100 gradient updates. This setup allows us to capture the full evolution of the hyperplanes and weight assignments throughout the training process.

**Training dynamics and convergence behavior.** Figure 6 presents the complete optimization trajectory of PCH on the same 2D triangular dataset used in Figure 2. This extended example visualizes how the three hyperplanes evolve throughout training, how their associated weight assignments stabilize.

The top row of Figure 6 shows the position and orientation of all three hyperplanes over 100 iterations, where lighter colors correspond to early iterations and darker colors correspond to later iterations. The hyperplanes move gradually from their random initial orientations toward the true supporting directions that define the convex boundary of the positive region. By iteration 30, the hyperplanes have already reached nearly their final geometry, and the remaining 70 iterations only produce very small adjustments. By the final iterations, each hyperplane has moved into a tight supporting position: all three hyperplanes pass through the corresponding positive support points and lie nearly flush against the outer boundary of the positive region. The resulting hyperplanes overlap almost perfectly with their limiting orientations, making the convergence of the geometric parameters visible.

The lower rows plot the evolution of the weight assignment for each hyperplane. Although weight assignments are conceptually defined over samples, PCH's formulation allows them to be visualized as a function of the signed distance $a_s^\top x - b_s$ because the weight of a sample depends on this signed distance within the surrogate. Thus, each curve shows: for all samples having a given signed distance to hyperplane $s$, what weight is assigned to them during that iteration. Initially, these curves are highly irregular and differ drastically across iterations. As optimization progresses, the curves rapidly collapse into stable profiles, indicating that the weight vector associated with each hyperplane has converged.

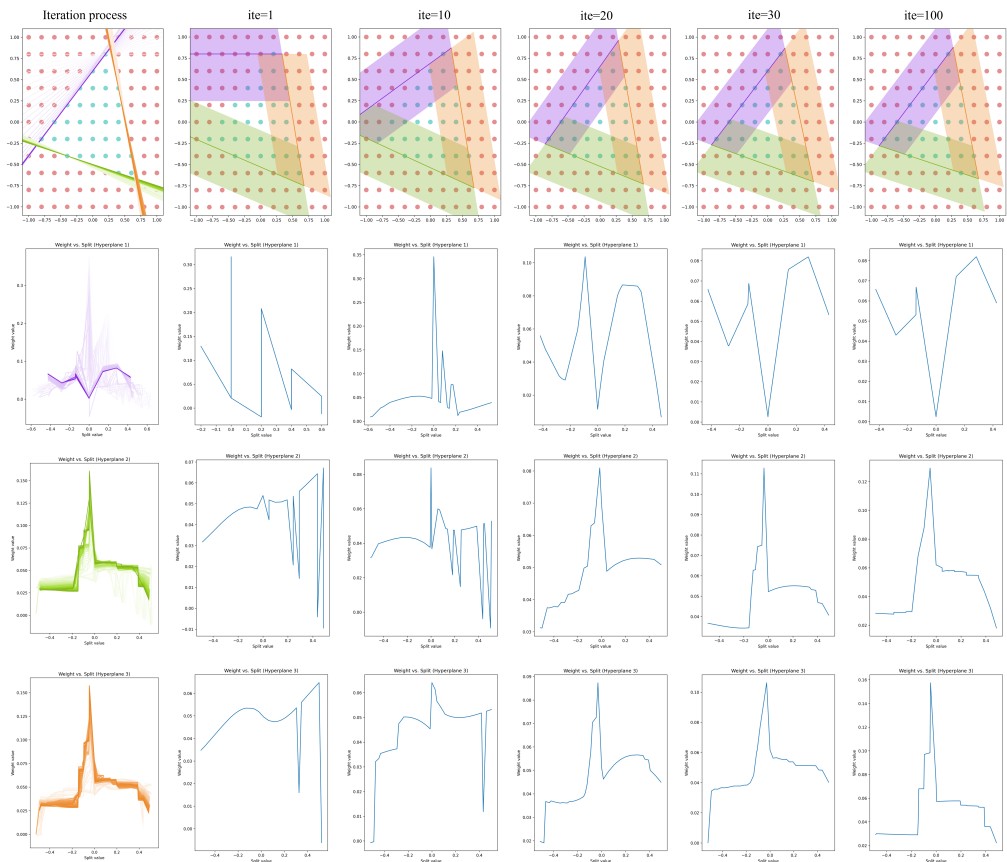

Figure 6: **Training dynamics of PCH on the 2D triangular dataset.**

**Robustness to random initialization.** Across six independent random initializations, all three hyperplanes converge to the close supporting configuration on the 2D triangular dataset. Despite large variations in the initial directions, the trajectories rapidly contract toward the correct boundary. The final configurations almost perfectly overlap and each hyperplane simultaneously passes through the corresponding positive support vectors, forming a tight outer envelope. This consistent convergence demonstrates that PCH is highly insensitive to initialization, providing strong empirical evidence that the optimization does not become trapped in poor local minima on this task.

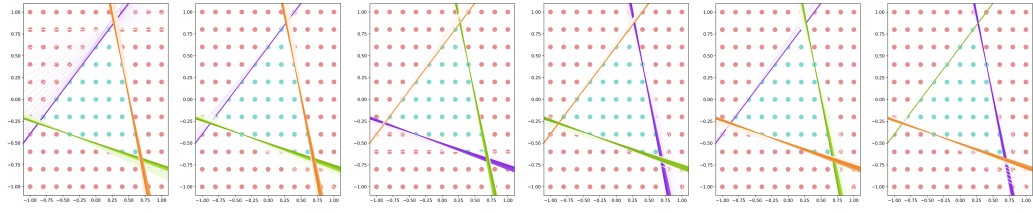

Figure 7: Hyperplane trajectories under random initializations.

**Relation to the MINLP formulation.** Figure 8 compares PCH with several representative baselines, and the MILP formulation in Equation 12. We do not compare the MINLP formulation in Equation 2 because the MINLP is extremely difficult to solve even in two dimensions. All baseline methods are run with their default hyperparameters. As shown in the figure, PCH is the only method whose learned hyperplanes lie tightly against the positive support points, producing a polyhedron

that is visually indistinguishable from the ideal supporting hull. In contrast, the MILP boundary leaves a clear gap between the hull and multiple positive support vectors, and geometric baselines (QH, DCH, PinCHD) introduce angular distortion or overshoot regions. DeepHull produces a smooth implicit boundary that cannot recover the polyhedral structure ($Ax \geq b$). These comparisons demonstrate that PCH yields the most geometrically faithful approximation to the underlying polyhedral trust region.

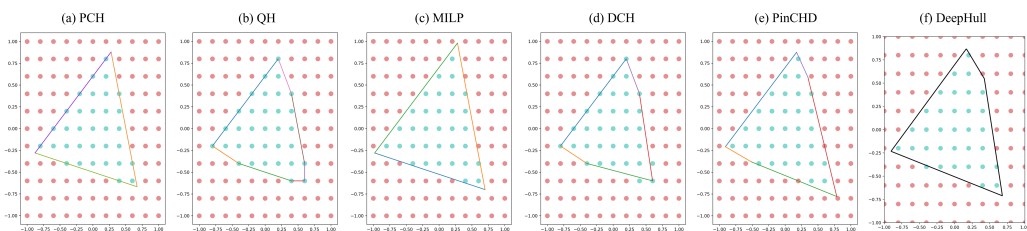

Figure 8: Comparison of PCH with MILP relaxation and representative geometric baselines on the 2D triangular dataset.

**Summary.** The triangular example provides a comprehensive view of how PCH operates in practice. The optimization trajectory shows that the surrogate updates drive the hyperplanes toward their tight supporting positions, while the weight assignments quickly stabilize into consistent profiles. The multi-start experiments further demonstrate that the procedure is highly insensitive to initialization, consistently converging to the same boundary. Finally, the comparison with the MILP relaxation and classical geometric baselines highlights that PCH yields the most faithful reconstruction of the underlying polyhedral trust region. These observations collectively illustrate the stability, robustness, and geometric accuracy of the proposed framework.

### G.3 POLYHEDRAL SEPARATION

This experiment evaluates the ability of convex hull learning methods to recover polyhedral decision regions composed of multiple supporting hyperplanes. We construct a family of synthetic datasets named PH, where the positive class forms a convex polyhedron in $\mathbb{R}^d$ bounded by $S = 15$ randomly generated hyperplanes.

**Dataset generation.** Each PH dataset contains 40,000 samples uniformly drawn from the hypercube $[-1, 1]^d$. To define a polyhedral region, we randomly generate $S = 15$ hyperplanes. For each hyperplane $j$, its normal vector $a^j$ is sampled from a uniform distribution over $[-1, 1]^d$. The bias $b^j$ is set as the 4th percentile of the projection values of all samples on $a^j$, ensuring that roughly 4% of the data lies below the hyperplane. A point is labeled positive if it satisfies all $S$ hyperplane constraints $a^{j\top} x \geq b^j$; otherwise, it is labeled negative. This results in a convex polytope enclosing approximately half of the samples, i.e., $(1 - 4\%)^{15} \approx 54\%$.

**Training settings.** We compare our method (PCH) with **QH** (Barber et al., 1996), **CGHC** (Barbato et al., 2024), **DCH** (Blum et al., 2019), **PinCHD** (Leng et al., 2019), and **DeepHull** (Balestriero et al., 2022). For all datasets, the number of hyperplanes is fixed at 15. For PCH, we use 15 outer iterations, each with 100 gradient descent steps, a learning rate of 0.1, and a loss smoothing temperature of $\beta = 4$. The margin shift coefficient $m_{\text{shift}}$ is set to 0.03, and the weight gradient update rate $\eta$ is set to 0.0001. CGHC is implemented following the main text of Barbato et al. (2024). The approximation rate of DCH is set to 0.05. PinCHD is implemented following the main text of Leng et al. (2019). The DeepHull model uses a single hidden layer with 7 neurons trained for 1500 epochs.

**Main results.** Table 6 and Table 7 provide a comprehensive comparison of six methods across increasing dimensions in polyhedral separation tasks. We evaluate their trust-region accuracy, along with training time and model complexity to assess scalability and computational efficiency.

PCH achieves **consistently high accuracy** (above 99% in all settings), **stable model complexity** (fixed at 15 hyperplanes), and **efficient training time** that scales gently with dimensionality. This demonstrates that PCH can effectively handle high-dimensional data without relying on combinatorial solvers or enumeration strategies, thanks to its continuous optimization formulation and strategic weight assignment. Even at 100 dimensions, PCH retains 99.95% accuracy with training time under 2 minutes, highlighting strong scalability and resilience to dimensional growth.

In contrast, CGHC experiences clear degradation as dimensionality increases. While it achieves competitive accuracy in low dimensions (99.71% at 2D), its performance drops to 57.20% at 100D. This decline is accompanied by substantial computational cost, with training time exceeding one hour at 100D. While CGHC attempts to mitigate combinatorial complexity via column generation, its reliance on MILP subproblems with binary variables still incurs a steep increase in computational burden as the problem dimension or sample size grows. The result is low accuracy, solver instability and early termination in high-dimensional cases.

QH achieves perfect trust-region accuracy in low dimensions (up to 8D), as it computes the exact convex hull enclosing all positive samples. However, it quickly becomes intractable as dimensionality increases. For instance, the number of output facets exceeds 23 million at just 8 dimensions, resulting in excessive memory usage and prohibitive runtime. Beyond 8D, QH fails to complete within a reasonable time budget, and results are omitted.

Compared with these, DCH, PinCHD, and DeepHull show mixed performance. DCH relies on sparse vertex selection without label supervision, achieving moderate accuracy in low dimensions but degrading rapidly as dimensionality increases, along with exploding model complexity. PinCHD performs consistently better, maintaining relatively high accuracy across dimensions, though its complexity grows with the number of iterations and selected anchors. DeepHull, while efficient due to its neural approximation, suffers from lower accuracy and lacks explicit linear constraints, which reduces interpretability and complicates its integration into downstream optimization tasks. For fairness of comparison in Table 7, we convert the learned neural boundary into its equivalent set of linear constraints and report the number of resulting constraints as the model complexity.

Overall, the results validate the core design goals of PCH: to provide scalable, structure-aware compact trust region modeling with strong performance guarantees in both low- and high-dimensional scenarios. Its consistent accuracy, computational efficiency, and controlled model complexity make it well-suited for practical deployment in large-scale trust region learning tasks.

Table 6: Comparison on polyhedral separation tasks. PCH maintains stable accuracy, efficiency and complexity across dimensions, while QH and CGHC deteriorate in high dimensions. A dash "–" indicates that QH fails to scale to the corresponding high-dimensional setting.

| Dimension | Problem | Trust-region Accuracy/% | | | Training Time/s | | | Model Complexity | | |
|---|---|---|---|---|---|---|---|---|---|---|
| | | QH | CGHC | PCH | QH | CGHC | PCH | QH | CGHC | PCH |
| | PH_15_2 | 100.00 | 99.71 | **100.00** | 0.02 | 599.81 | 73.43 | 37 | 15 | **7** |
| | PH_15_3 | 100.00 | 98.74 | **100.00** | 0.00 | 652.00 | 80.31 | 532 | 15 | **12** |
| | PH_15_4 | 100.00 | 96.45 | **100.00** | 0.02 | 698.43 | 85.07 | 4660 | 15 | **15** |
| Low | PH_15_5 | 100.00 | 92.19 | 99.52 | 0.22 | 497.41 | 89.74 | 40700 | 15 | **15** |
| | PH_15_6 | 100.00 | 92.68 | **100.00** | 4.15 | 717.34 | 90.19 | 336283 | 15 | **15** |
| | PH_15_7 | 100.00 | 86.06 | 99.67 | 56.84 | 407.45 | 91.48 | 2.66E+6 | 15 | **15** |
| | PH_15_8 | 100.00 | 87.59 | **100.00** | 881.88 | 922.55 | **91.94** | 2.30E+7 | 15 | **15** |
| High | PH_15_10 | – | 82.26 | **100.00** | – | 919.31 | **93.34** | – | 15 | **15** |
| | PH_15_100 | – | 57.20 | **99.95** | – | 13537.29 | **117.32** | – | 15 | **15** |

**Training dynamics.** Figure 9 shows the evolution of trust-region accuracy and cumulative training time *against* outer iteration count for different dimensional settings. PCH exhibits stable and steady convergence across all dimensions, progressively improving its accuracy over iterations without significant fluctuations. In contrast, CGHC frequently struggles to reach comparable accuracy under the same hyperplane budget. Notably, in several high-dimensional cases, CGHC terminates prematurely before completing the full 15 iterations. This is due to its reliance on dual variable updates and KKT-based termination conditions, which can be sensitive to solver precision and may incorrectly trigger early stopping when the underlying MILP problem is defined on high-dimensional

Table 7: More comparison on polyhedral separation tasks. A dash "–" indicates that models fail to scale to the corresponding high-dimensional setting.

| Dimension | Problem | Trust-region Accuracy/% | | | Training Time/s | | | Model Complexity | | |
|---|---|---|---|---|---|---|---|---|---|---|
| | | DCH | PinCHD | DeepHull | DCH | PinCHD | DeepHull | DCH | PinCHD | DeepHull |
| Low | PH_15_2 | 97.88 | 99.80 | 97.01 | 0.74 | 37.06 | 4.26 | 9 | 11 | 15 |
| | PH_15_3 | 96.58 | 97.34 | 92.62 | 7.50 | 63.39 | 4.08 | 188 | 38 | 15 |
| | PH_15_4 | 94.93 | 95.85 | 86.31 | 51.51 | 92.56 | 4.05 | 2.96E+3 | 91 | 15 |
| | PH_15_5 | 88.38 | 95.84 | 86.62 | 77.62 | 135.98 | 4.07 | 2.52E+4 | 152 | 15 |
| | PH_15_6 | 76.75 | 94.35 | 78.54 | 81.11 | 167.70 | 4.04 | 1.70E+5 | 250 | 15 |
| | PH_15_7 | 62.83 | 94.62 | 82.33 | 94.12 | 338.96 | 4.14 | 1.05E+6 | 351 | 15 |
| | PH_15_8 | 45.26 | 95.26 | 82.01 | 185.12 | 362.87 | 4.00 | 6.18E+6 | 480 | 15 |
| High | PH_15_10 | – | 96.11 | 82.36 | – | 495.72 | 4.00 | – | 880 | 15 |
| | PH_15_100 | – | 57.95 | 64.39 | – | 8741.04 | 5.08 | – | 1.50E+3 | 15 |

and large-scale datasets. These observations further highlight the robustness and stability of PCH in optimization dynamics.

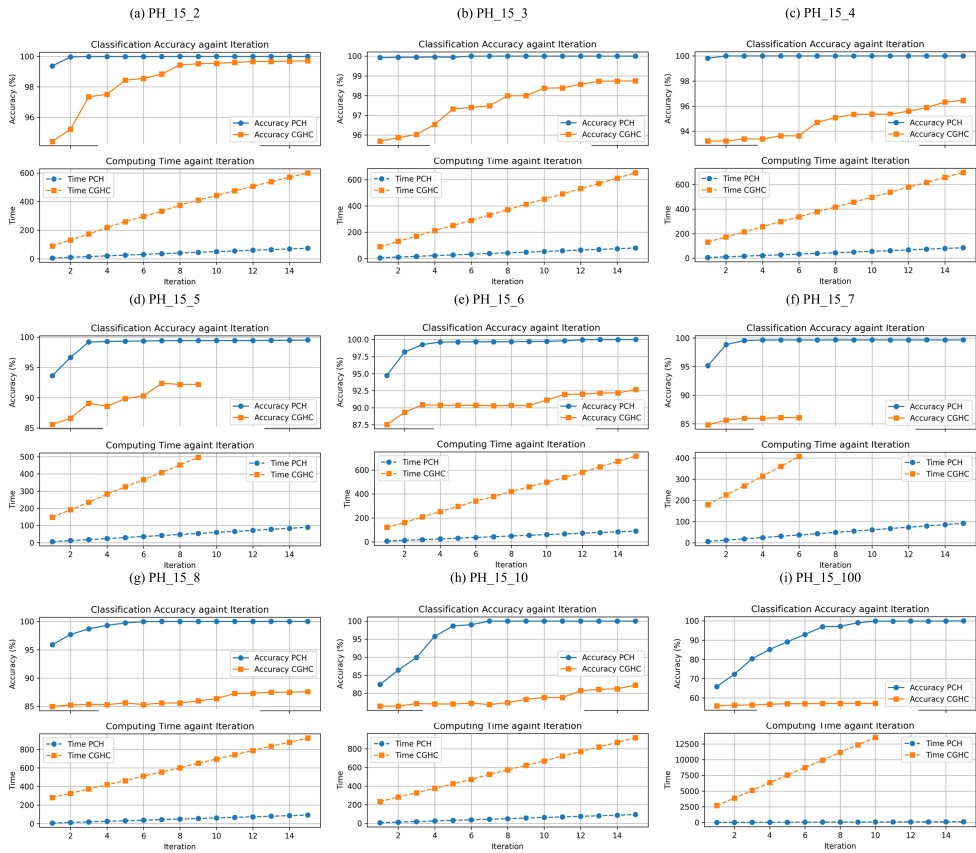

Figure 9: Trust-region accuracy and time against iteration for PCH and CGHC across dimensions 2 to 100.

**Visualization in 2D.** We visualize the decision boundaries on the 2D PH dataset in Figure 10. PCH is able to reconstruct a tight polyhedron that encloses all positive samples and excludes nearly all negatives, while CGHC and other methods show slightly looser boundaries.

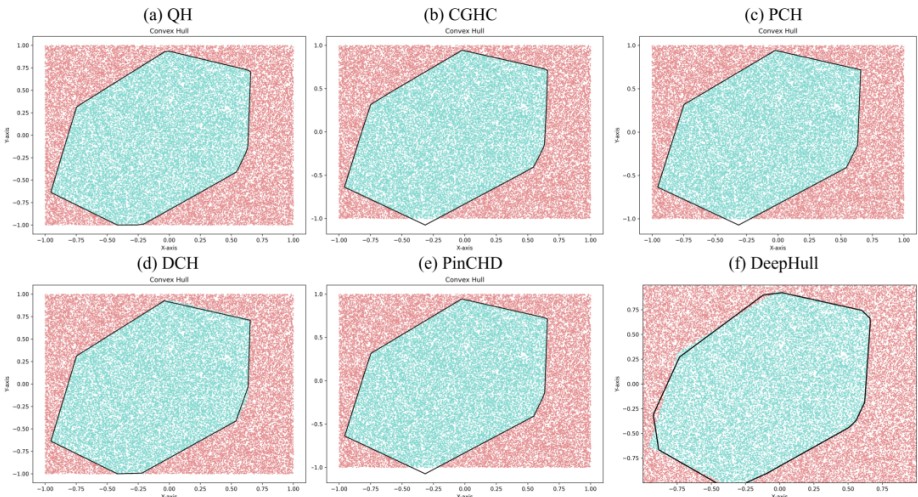

Figure 10: Learned convex hulls on 2D polyhedral dataset (PH_15_2). Blue: positive samples; red: negative samples.

**Robustness to noise injection.** To assess robustness under label noise, we construct noisy variants of the polyhedral datasets based on the same ground-truth polyhedra. For each dimension, we first generate the clean dataset as described above. We then draw an additional 2,000 points uniformly from $[-1, 1]^d$ and label them according to the same convex hull. Among those labeled as positive, we flip a subset to negative if they lie within a small margin band of width 0.1 around at least one supporting hyperplane. All methods are trained with exactly the same hyperparameters as in the clean polyhedral experiments.

Tables 8 and 9 report the trust-region accuracy, training time, and model complexity on the noisy polyhedral datasets. Across all dimensions, PCH maintains very high accuracy (typically around 99%) with a fixed budget of 15 hyperplanes and training time on the order of one to two minutes. Among the baselines, DeepHull is relatively more sensitive to noise. These results indicate that PCH not only scales well in the clean polyhedral setting, but also preserves tight and reliable trust regions in the presence of substantial interior noise.

Table 8: Comparison on noisy polyhedral separation tasks. PCH maintains stable accuracy, efficiency and complexity across dimensions, while QH and CGHC deteriorate in high dimensions. A dash "–" indicates that QH fails to scale to the corresponding high-dimensional setting.

| Dimension | Problem | Trust-region Accuracy/% | | | Training Time/s | | | Model Complexity | | |
|---|---|---|---|---|---|---|---|---|---|---|
| | | QH | CGHC | PCH | QH | CGHC | PCH | QH | CGHC | PCH |
| Low | PH_15_2_N | 99.41 | 99.08 | 99.38 | 0.02 | 698.74 | 89.67 | 37 | 15 | **15** |
| | PH_15_3_N | 99.39 | 97.10 | 99.35 | 0.01 | 760.21 | 102.33 | 534 | 15 | **15** |
| | PH_15_4_N | 99.37 | 94.87 | 99.33 | 0.02 | 544.23 | 93.29 | 4.69E+3 | 15 | **15** |
| | PH_15_5_N | 99.44 | 93.52 | 99.32 | 0.23 | 610.57 | 93.25 | 4.11E+4 | 15 | **15** |
| | PH_15_6_N | 99.34 | 93.09 | 95.82 | 4.20 | 569.94 | 91.77 | 3.39E+5 | 15 | **15** |
| | PH_15_7_N | 99.31 | 89.42 | 99.24 | 59.67 | 673.50 | 93.86 | 2.70E+6 | 15 | **15** |
| | PH_15_8_N | 99.4 | 89.50 | 99.32 | 928.50 | 601.10 | **91.90** | 2.35E+7 | 15 | **15** |
| High | PH_15_10_N | – | 78.39 | **98.05** | – | 779.69 | **94.03** | – | 15 | **15** |
| | PH_15_100_N | – | 57.71 | **98.93** | – | 19825.01 | **118.16** | – | 15 | **15** |

Figure 11 visualizes the learned convex hulls on the noisy 2D dataset. Interior noisy negatives appear as scattered red points inside the positive region. All methods are necessarily forced to include some of these points, but there are clear differences in how tightly they follow the outer envelope of the positive region. PCH produces a boundary that closely tracks the true outer shape while using the prescribed number of hyperplanes. QH resembles the ground-truth hull without considering any negative points, whereas CGHC, DCH, PinCHD, and DeepHull tend to generate slightly looser or more distorted hulls. This 2D visualization is consistent with the quantitative results: PCH preserves

Table 9: More comparison on noisy polyhedral separation tasks. A dash "–" indicates that models fail to scale to the corresponding high-dimensional setting.

| Dimension | Problem | Trust-region Accuracy/% | | | Training Time/s | | | Model Complexity | | |
|-----------|---------|------|--------|----------|------|--------|----------|------|--------|----------|
| | | DCH | PinCHD | DeepHull | DCH | PinCHD | DeepHull | DCH | PinCHD | DeepHull |
| | PH_15_2_N | 97.60 | 99.08 | 97.03 | 0.87 | 53.29 | 4.61 | 8 | 11 | 15 |
| | PH_15_3_N | 95.97 | 97.13 | 92.67 | 8.84 | 94.44 | 4.66 | 180 | 37 | 15 |
| | PH_15_4_N | 94.54 | 95.01 | 87.61 | 52.69 | 111.83 | 4.33 | 2.94E+3 | 93 | 15 |
| Low | PH_15_5_N | 88.20 | 94.66 | 85.14 | 80.70 | 167.53 | 4.31 | 2.52E+4 | 144 | 15 |
| | PH_15_6_N | 76.93 | 94.00 | 79.69 | 83.03 | 363.43 | 4.41 | 1.71E+5 | 235 | 15 |
| | PH_15_7_N | 63.15 | 94.19 | 80.15 | 91.73 | 408.46 | 4.45 | 1.05E+6 | 340 | 15 |
| | PH_15_8_N | 45.88 | 94.86 | 78.52 | 184.57 | 495.74 | 4.13 | 6.18E+6 | 494 | 15 |
| High | PH_15_10_N | – | 95.35 | 76.97 | – | 561.35 | 4.28 | – | 1.02E+3 | 15 |
| | PH_15_100_N | – | 57.15 | 74.39 | – | 12254.17 | 5.19 | – | 1.50E+3 | 15 |

a tight outer approximation of the positive class even when noisy negatives are injected inside the positive region.

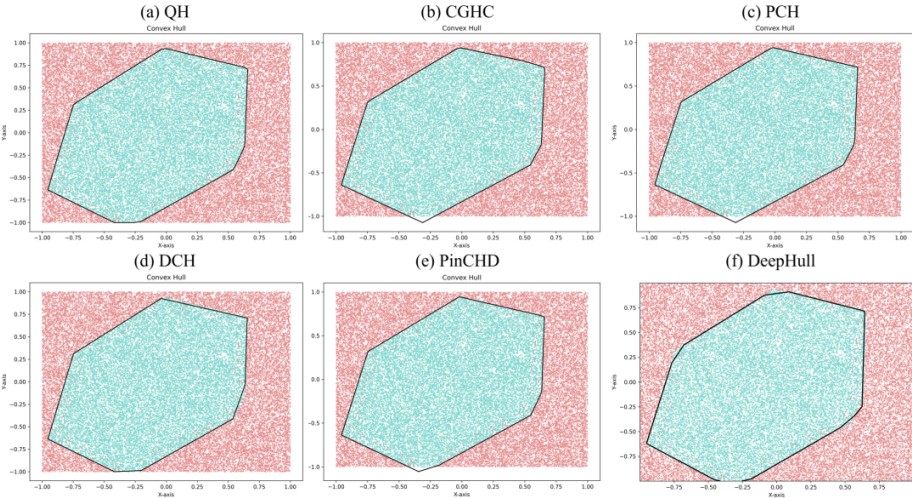

Figure 11: Learned convex hulls on the noisy 2D polyhedral dataset (PH_15_2_N).

**Summary.** This experiment demonstrates that our method achieves superior scalability, efficiency, and robustness in learning polyhedral trust regions, particularly in high-dimensional and large-scale settings.

### G.4 NONLINEAR CONVEX SEPARATION

**Dataset construction.** Each NLC dataset contains 40,000 samples uniformly drawn from the hypercube $[-1, 1]^d$. To construct a smooth convex region, we compute the Euclidean norm of each sample from the origin and use the median norm value as a threshold. Samples whose norm is below the threshold are labeled as positive, while the remaining samples are labeled negative. This produces a balanced dataset where the positive class forms a hypersphere centered at the origin. The resulting boundary is convex but cannot be exactly represented by a small number of linear facets, making it a challenging testbed for learning trust regions with convex hulls.

**Training settings.** To ensure a fair comparison, we use the same experimental settings as in the polyhedral case, except for the hyperplane budget, which is adjusted to scale with input dimension as $\lceil 15 \cdot \sqrt{d-1} \rceil$. This scaling strategy provides flexibility to approximate curved convex boundaries in high dimensions, while keeping model size manageable and comparable across methods.

**Main results.** Table 10 and Table 11 summarize the results on NLC datasets across different dimensions. PCH consistently achieves strong trust-region accuracy while maintaining moderate training time and controlled model complexity. Unlike methods that aim to approximate the ground-truth curved boundary, our goal is to learn a compact and tight trust region that encloses all positive samples while excluding as many negative samples as possible, under a limited hyperplane budget. The results show that PCH successfully constructs trust regions across all tested dimensions. For instance, at 100 dimensions, it achieves 99.01% accuracy with 150 hyperplanes and a training time of 1747 seconds, demonstrating both scalability and robustness.

Table 10: Performance comparison on nonlinear convex (NLC) datasets of increasing dimensionality. A dash "–" indicates that QH fails to scale to the corresponding high-dimensional setting.

| Dimension | Problem | Trust-region Accuracy/% | | | Training Time/s | | | Model Complexity | | |
|---|---|---|---|---|---|---|---|---|---|---|
| | | QH | CGHC | PCH | QH | CGHC | PCH | QH | CGHC | PCH |
| Low | NLC_N_2 | 100.00 | 99.27 | 99.40 | 0.02 | 629.96 | 77.51 | 93 | 15 | **15** |
| | NLC_N_3 | 100.00 | 93.09 | 94.35 | 0.01 | 725.49 | 111.84 | 1.28E+3 | 22 | **22** |
| | NLC_N_4 | 100.00 | 86.11 | 89.55 | 0.04 | 757.84 | 122.59 | 1.13E+4 | 26 | **26** |
| | NLC_N_5 | 100.00 | 76.70 | 84.61 | 0.55 | 916.29 | 148.18 | 9.34E+4 | 30 | **30** |
| | NLC_N_6 | 100.00 | 71.60 | 79.19 | 8.63 | 1145.53 | 166.30 | 7.23E+5 | 34 | **34** |
| | NLC_N_7 | 100.00 | 66.62 | 75.11 | 111.03 | 1350.59 | 187.42 | 5.62E+6 | 37 | **37** |
| | NLC_N_8 | 100.00 | 60.38 | 71.77 | 1701.16 | 8804.44 | **192.31** | 4.50E+7 | 40 | **40** |
| High | NLC_N_10 | – | 57.60 | **69.23** | – | 7705.48 | **235.14** | – | 45 | **45** |
| | NLC_N_100 | – | 68.89 | **99.01** | – | 30762.99 | **1747.66** | – | 150 | **150** |

Table 11: Performance comparison on nonlinear convex (NLC) datasets of increasing dimensionality. A dash "–" indicates that models fail to scale to the corresponding high-dimensional setting.

| Dimension | Problem | Trust-region Accuracy/% | | | Training time/s | | | Model Complexity | | |
|---|---|---|---|---|---|---|---|---|---|---|
| | | DCH | PinCHD | DeepHull | DCH | PinCHD | DeepHull | DCH | PinCHD | DeepHull |
| Low | NLC_N_2 | 95.24 | 99.41 | 97.64 | 0.94 | 26.97 | 4.23 | 15 | 24 | 15 |
| | NLC_N_3 | 91.01 | 99.06 | 93.79 | 6.88 | 40.23 | 3.75 | 242 | 125 | 21 |
| | NLC_N_4 | 86.02 | 99.05 | 92.61 | 35.45 | 60.48 | 3.90 | 4.39E+3 | 309 | 25 |
| | NLC_N_5 | 74.05 | 99.39 | 88.23 | 76.77 | 92.94 | 4.19 | 4.05E+4 | 632 | 29 |
| | NLC_N_6 | 54.69 | 99.76 | 88.11 | 80.97 | 112.76 | 3.63 | 1.98E+5 | 1.01E+3 | 33 |
| | NLC_N_7 | 36.48 | 99.96 | 86.31 | 93.50 | 238.47 | 4.24 | 1.03E+6 | 1.48E+3 | 37 |
| | NLC_N_8 | 22.04 | 98.22 | 86.75 | 172.37 | 266.35 | 4.20 | 5.59E+6 | 1.50E+3 | 39 |
| High | NLC_N_10 | – | 92.46 | 85.99 | – | 371.34 | 4.96 | – | 1.50E+3 | 45 |
| | NLC_N_100 | – | 53.75 | 75.39 | – | 8542.03 | 9.70 | – | 1.50E+3 | 149 |

In contrast, CGHC experiences clear degradation as dimensionality increases. While it achieves competitive accuracy in low dimensions (e.g., 99.27% at 2D), its performance drops significantly to 68.89% at 100D. This decline is accompanied by substantial computational cost, with training time exceeding 30,000 seconds. Although CGHC leverages a column generation strategy to heuristically solve its MILP formulation, the underlying MILP formulation still introduces a large number of binary decision variables. As dimension and sample size grow, this leads to severe combinatorial complexity, which hinders its scalability in high-dimensional settings.

QH produces perfect separation accuracy for all low-dimensional cases, since it computes the exact convex hull of the positive samples. However, it becomes computationally intractable beyond 8D due to exponential growth in output hyperplanes. As a result, QH fails to complete within a reasonable time budget, and results are omitted.

Compared with these methods, DCH, PinCHD, and DeepHull exhibit clear limitations in the trust region setting. DCH selects sparse representative vertices and internally approximates the convex hull, but since it does not enforce full positive inclusion, its accuracy rapidly degrades as dimension grows and its model complexity explodes. PinCHD provides stronger separation performance through projection-based initialization and iterative refinement, yet it cannot control the number of hyperplanes, leading to unbounded model growth in higher dimensions. DeepHull leverages neural approximations to capture convex regions with low training time, but it does not produce explicit linear constraints and similarly fails to guarantee inclusion of all positives, limiting its interpretability and downstream usability.

These findings confirm that PCH effectively supports trust region learning with convex hulls, even when the ground-truth boundary is nonlinear. It maintains strong performance in terms of both trust-region accuracy and computational efficiency, making it well suited for large-scale, high-dimensional applications that require compact and adaptive trust region modeling.

**Training dynamics.** Figure 12 shows the evolution of trust-region accuracy and runtime for CGHC and PCH. The results are similar to the results in the polyhedral datasets. PCH also exhibits steady and monotonic improvement in accuracy compared with CGHC, and its runtime scales smoothly across iterations.

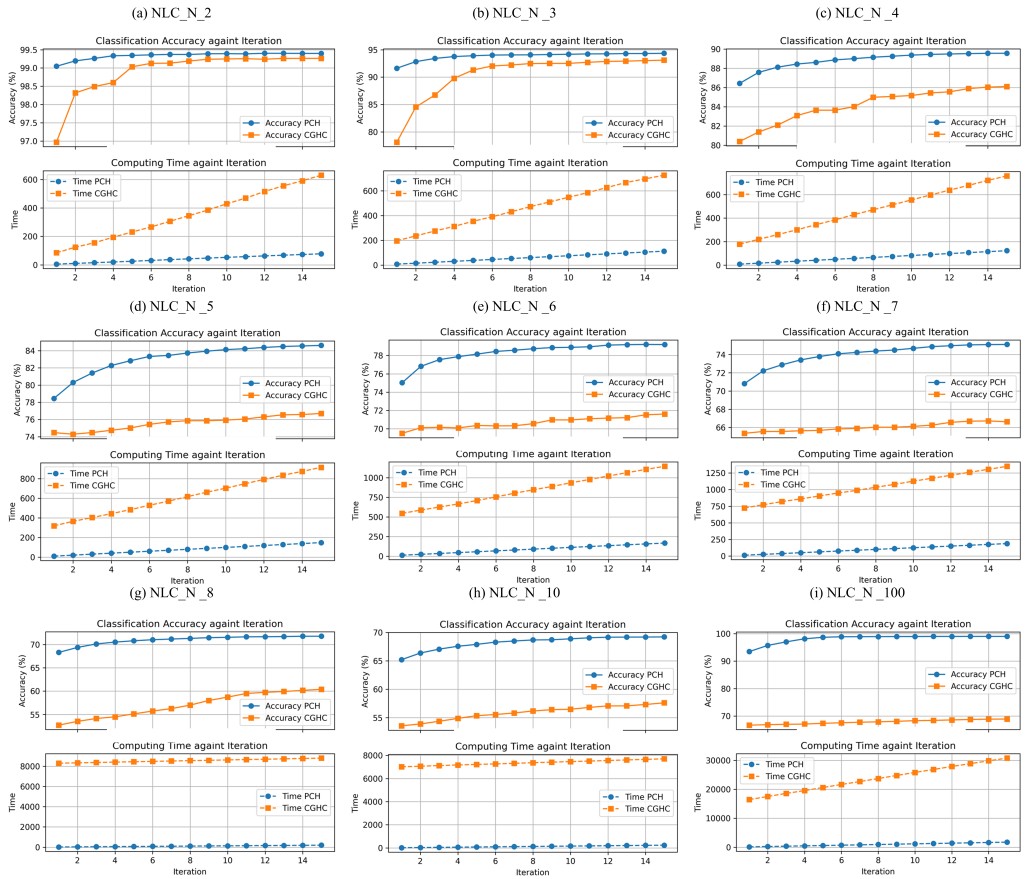

Figure 12: Trust-region accuracy and runtime against iteration on NLC datasets across dimensions.

**Geometric visualization.** To further illustrate model behavior, Figure 13 visualizes the learned convex hulls in 2D. QH constructs a perfectly tight boundary by computing the exact convex hull of the positive class with excessive hyperplanes. Both PCH and CGHC yield similar trust regions that closely follow the curved boundary, despite using only 15 supporting hyperplanes. Other methods also achieve similar results without controlling supporting hyperplanes. This confirms that in low-dimensional convex tasks, all methods can construct an effective convex hull for trust region modeling, with PCH offering a more structured optimization-based formulation.

**Scalability trends.** Figure 14 summarizes performance trends over increasing dimensions. Accuracy (left), training time (middle), and model complexity (right) are compared. PCH retains competitive accuracy and model complexity, whereas CGHC suffers from an accuracy drop and exponential computational cost. QH remains exact but becomes infeasible after 8D.

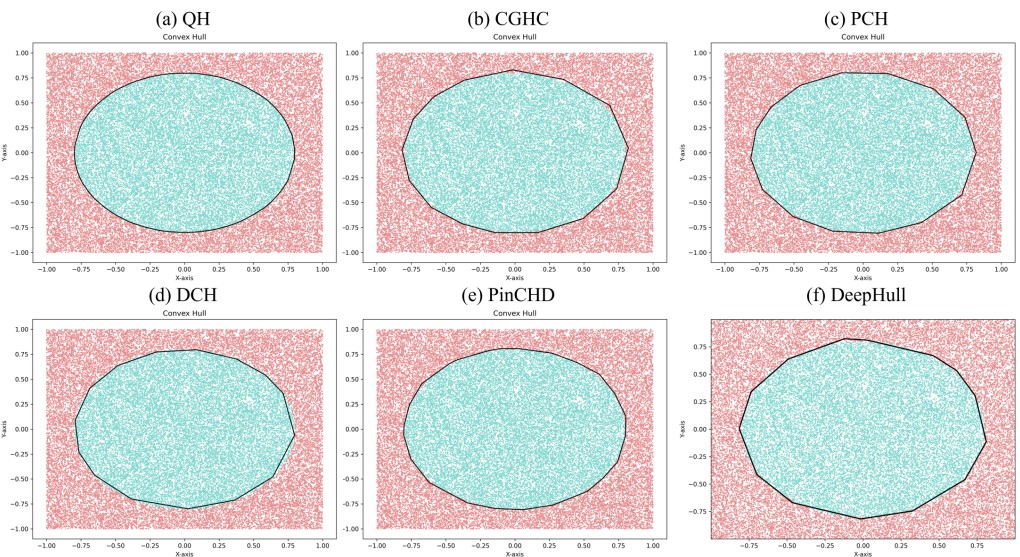

Figure 13: Visual comparison of 2D NLC convex hulls.

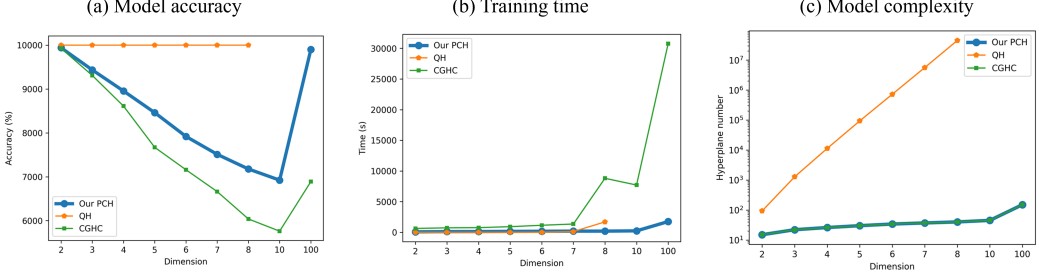

Figure 14: Trends of accuracy, training time, and model complexity against dimensions for NLC datasets.

**The effect of the hyperplane budget.**    To further examine how the hyperplane budget $S$ influences the quality of the learned convex separator, we consider the 2D circular dataset. Because the target set is not polyhedral, any convex-hull based approach necessarily produces a polygonal outer approximation whose tightness depends on the number of available hyperplanes. Figure 15 illustrates the PCH solutions obtained with $S$ ranging from 4 to 15.

When $S$ is small (e.g., $S=4$–6), the resulting polyhedron is a coarse envelope that captures the global geometry but cannot closely follow the curved boundary. As $S$ increases, the learned polytope becomes progressively sharper and more circular: additional hyperplanes allow the model to place supporting facets at finer angular resolutions, reducing the discrepancy between the polygon and the underlying smooth circular shape. By $S=12$–15, the separator visually aligns with the circular boundary and achieves high trust-region accuracy, matching the expected behavior of polyhedral approximations to nonlinear convex sets.

This experiment highlights how PCH adapts naturally to nonlinear convex regions: the method always returns a valid convex outer approximation, and the attainable approximation is controlled directly by the hyperplane budget. Larger values of $S$ enable tighter geometric fidelity without compromising the convexity or the interpretability of the resulting model.

**Summary.**    This experiment confirms that PCH generalizes effectively to nonlinear convex separation tasks. It achieves accurate and geometrically compact separation under a limited hyperplane budget, maintains competitive performance even in high-dimensional settings, and avoids the computational burden of combinatorial solvers. These results highlight the flexibility and scalability of PCH for large-scale trust region modeling with polyhedral convex hull learning.

### G.5    Nonlinear nonconvex trust region learning

**Dataset Generation.**    To evaluate the robustness of our method in handling the highly irregular, nonconvex trust region, we construct a two-dimensional synthetic dataset inspired by the transient stability boundaries in power systems. The positive region is defined as the union of two components. The first is a parabolic head consisting of points $(x^{(1)}, x^{(2)})$ with $x^{(1)} \geq 8$ and $x^{(2)} \leq -0.1(x^{(1)}-8)^2+18$. The second is a circular body defined by $(x^{(1)}-8)^2+(x^{(2)}-12)^2 \leq 36$ for $x^{(1)} < 8$. To introduce nonconvexity, a lower parabolic constraint $x^{(2)} \geq -0.1(x^{(1)}-12)^2+8$ is further imposed on the entire region. This compound structure forms a drop-like shape as shown in Figure 16. Points within this nonconvex region are labeled positive; others are labeled negative. We generate 10,000 samples uniformly over the $[0, 20]^2$ grid to construct the dataset.

Unlike typical classification settings that tolerate exclusion of a few positive samples to gain broader separation from negatives, trust region modeling emphasizes full inclusion of the positive class while excluding as many negatives as possible. This distinction is critical: in standard classification, misclassifying a few positive samples can be acceptable if it helps correctly separate many negatives. In contrast, trust region learning requires full inclusion of feasible examples, as excluding even a few may compromise the reliability of downstream decisions. Therefore, learning compact convex hulls that tightly enclose all positive samples enables safer, bias-resistant modeling in subsequent classification and constraint learning tasks.

**Main Results.**    Table 12 reports the performance of all methods on this nonconvex setting. PCH achieves high classification accuracy (92.10%) while maintaining a compact model with only 15 hyperplanes. Compared to CGHC (91.83%), PCH provides better geometric alignment and lower training time (25.85s vs. 615.35s). QH achieves slightly higher accuracy (92.15%) by tightly enclosing all positive samples, but at the cost of a significantly larger number of facets (27), and lacks flexibility in controlling model complexity. These results highlight the advantage of PCH in learning compact trust regions with controllable structure, even under nonconvex ground-truth boundaries.

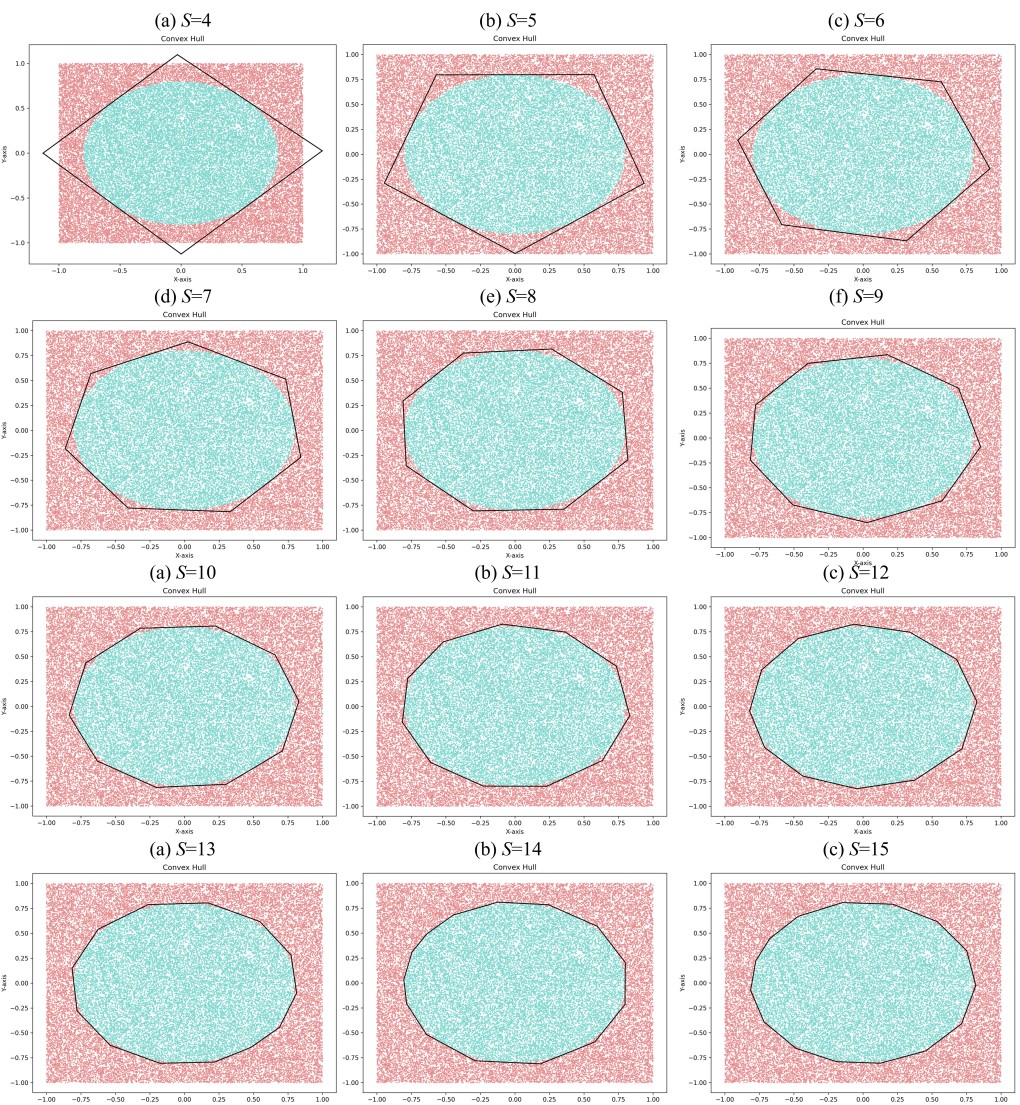

Figure 15: Effect of the hyperplane budget $S$ on 2D nonlinear convex separation.

Table 12: Performance comparison on nonconvex dataset (NLNC_N_2)

| Method | Accuracy (%) | Runtime (s) | Model Complexity |
|---|---|---|---|
| PCH | 92.10 | 25.85 | 15 |
| QH | 92.15 | 0.00 | 27 |
| CGHC | 91.83 | 615.34 | 15 |
| DCH | 91.87 | 2.04 | 27 |
| PinCHD | 91.86 | 10.78 | 14 |
| DeepHull | 87.05 | 3.02 | 15 |

**Visualization.**  Figure 16 compares the learned trust regions from different methods. QH recovers the exact convex hull of the positive class, closely following the outer envelope of the true nonconvex shape. PCH approximates the region using a tight piecewise-linear surface with 15 facets, successfully including all positives while excluding most negatives outside $\mathrm{conv}(\mathcal{D}_+)$. CGHC produces a noisier and less regular polyhedron under the same hyperplane budget, leading to reduced accuracy. DCH internally approximates QH and thus closely resembles the exact convex hull in low dimensions, but it does not enforce positive inclusion in general. PinCHD also yields a reasonable approximation, but without explicit control of the number of hyperplanes, its structure may become overly complex in higher dimensions. DeepHull generates a smooth neural approximation but lacks explicit facet representation and positive inclusion; as a result, its boundary drifts inward and fails to cover part of the positive region, leaving gaps in the trust region.

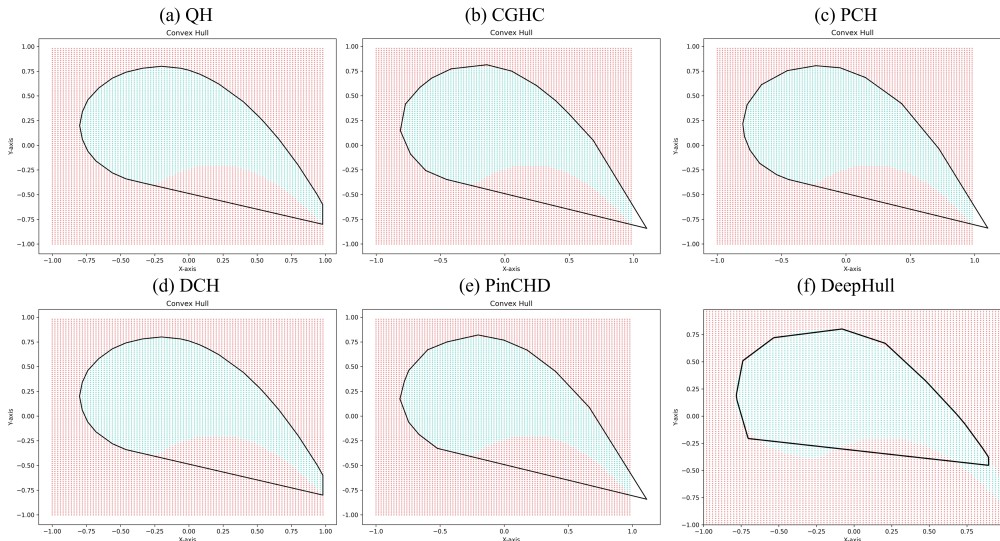

Figure 16: Visualization of convex hulls on NLNC dataset.

**Summary.**  This experiment shows that PCH remains effective even when the true positive region is nonconvex. While convex polytopes cannot fully represent nonconvex geometry, PCH succeeds in learning compact trust regions that enclose all positive samples while excluding as many negatives as possible. This highlights its practical utility in trust region modeling, where full positive coverage is essential to improve model performance in downstream tasks. The results suggest that PCH can serve as a reliable tool for safe region construction in applications such as power system stability assessment and constraint learning, where strict adherence to feasible boundaries is critical.

### G.6  REAL-WORLD TRUST REGION LEARNING

**Datasets.**  We further evaluate PCH on a broad set of real-world datasets, including four tabular benchmarks (Breast Cancer, Spambase, Bace, HIV) and four more complex tasks (MNIST-35,

FMNIST-24, CIFAR-35, Adult Income). The image datasets are converted to binary classification problems by restricting the labels to class pairs 3 vs. 5 (MNIST-35), 2 vs. 4 (FMNIST-24), and 3 vs. 5 (CIFAR-35). Table 13 summarizes the sample sizes, feature dimensions, and hyperparameters used for training.

Table 13: Summary of real-world datasets and training settings.

| Dataset | Breast Cancer | Spambase | Bace | HIV | MNIST-35 | FMNIST-24 | CIFAR-35 | Adult Income |
|---|---|---|---|---|---|---|---|---|
| Number of samples ($n$) | 569 | 4,601 | 1,513 | 41,127 | 13,454 | 14,000 | 12,000 | 48,842 |
| Feature dimension ($d$) | 30 | 57 | 2,048 | 2,048 | 784 | 784 | 3,072 | 105 |
| Hyperplane budget ($S$) | 5 | 10 | 5 | 15 | 10 | 10 | 100 | 500 |
| Split temperature ($\beta$) | 12 | 16 | 4 | 4 | 4 | 4 | 4 | 4 |
| Learning rate | 0.01 | 0.01 | 0.001 | 0.001 | 0.001 | 0.001 | 0.001 | 0.01 |

**Main results.** Table 14 reports the trust-region accuracy, training time, and model complexity. Across all datasets, PCH attains the highest accuracy while maintaining a compact model size and moderate training time. On Breast Cancer, all methods reach near-perfect accuracy, and PCH matches the best result with only 5 hyperplanes. On Spambase, PCH substantially outperforms CGHC (94.41% vs. 86.70%) while keeping the hyperplane budget at 10 and the runtime below 3 seconds. For the high-dimensional Bace dataset, PCH achieves the best accuracy (98.61%) with 5 hyperplanes, whereas CGHC requires over 8000 seconds and PinCHD expands to 149 hyperplanes.

On the larger and more challenging datasets HIV, MNIST-35, FMNIST-24, CIFAR-35, and Adult Income, PCH scales successfully to tens of thousands of samples and up to 3072 features, handling class imbalance and complex data manifolds, while CGHC and PinCHD fail to produce results within a reasonable time. We emphasize that no parallel or specialized hardware implementation is used for PCH in these experiments; the reported runtimes therefore serve as a conservative indication of the method's scalability.

Table 14: Performance comparison on real-world datasets. Reported are trust-region accuracy (%), training time (s), and model complexity (number of hyperplanes). A dash "–" indicates that a method failed to scale to the dataset.

| Problem | Trust-region Accuracy/% | | | Training Time/s | | | Model Complexity | | |
|---|---|---|---|---|---|---|---|---|---|
| | CGHC | PinCHD | PCH | CGHC | PinCHD | PCH | CGHC | PinCHD | PCH |
| Breast Cancer | 99.82 | 99.82 | **99.82** | 33.00 | 0.19 | 2.55 | 5 | 10 | **5** |
| Spambase | 86.70 | 91.41 | **94.41** | 446.96 | 9.87 | **2.58** | 10 | 121 | **10** |
| Bace | 98.35 | 76.80 | **98.61** | 8075.50 | 257.65 | **155.53** | 5 | 149 | **5** |
| HIV | – | – | **99.94** | – | – | 1800.67 | – | – | **15** |
| MNIST-35 | – | – | **99.98** | – | – | 390.80 | – | – | **10** |
| FMNIST-24 | – | – | **99.10** | – | – | 1094.22 | – | – | **30** |
| CIFAR-35 | – | – | **92.58** | – | – | 26868.54 | – | – | **100** |
| Adult Income | – | – | **89.08** | – | – | 4044.94 | – | – | **500** |

We further provide a complementary comparison with DeepHull on the more complex datasets (MNIST-35, FMNIST-24, CIFAR-35, and Adult Income). As DeepHull is a neural convex-approximation model, we focus here on trust-region accuracy and the fraction of positive samples contained inside the learned region. The results in Table 15 show that PCH consistently achieves higher trust-region accuracy and, importantly, guarantees full inclusion of all positive samples (100% positive coverage on all four tasks), whereas DeepHull leaves a non-negligible portion of positives outside its learned region.

### G.7 SELECTIVE CLASSIFICATION

**Datasets.** To evaluate the performance of our trust region modeling approach, we conduct experiments on five binary classification datasets. The first dataset, NLNC_N_2, approximates the transient stability boundary of a power system, and its generation method was described earlier. The remaining datasets are well-established benchmarks for classification tasks: Breast Cancer, Spambase, Bace, and HIV.

Table 15: Comparison between PCH and DeepHull in trust-region learning. Reported are trust-region accuracy (%) and positive inclusion (% of positives inside the learned region).

| Problem | Trust-region Accuracy/% | | Positive Inclusion/% | |
|---|---|---|---|---|
| | DeepHull | PCH | DeepHull | PCH |
| MNIST-35 | 99.84 | **99.98** | 99.94 | **100** |
| FMNIST-24 | 91.84 | **99.10** | 96.43 | **100** |
| CIFAR-35 | 68.66 | **92.58** | 85.12 | **100** |
| Adult Income | 64.66 | **89.08** | 74.10 | **100** |

**Experimental Setup.** Each dataset is split into training and testing sets with a 4:1 ratio, ensuring that the testing set is kept unseen during the training process. We compare the performance of our proposed method (PCH), with that of CGHC for the task of selective classification. The QH method is excluded from the comparison due to its limitations in handling high-dimensional problems and its inflexibility in controlling the number of supporting hyperplanes.

For downstream tasks, we use two classifiers: CART (Classification and Regression Trees) and MLP (Multi-layer Perceptron). We employ default parameters for the CART model. The hyperparameters of the MLP classifier are selected based on training performance.

We evaluate each classification model (CART and MLP) under five configurations. The baseline configuration, denoted as "–", uses the classifier trained on the full dataset without trust region filtering. Configurations marked with $\dagger$ use the same classifier but apply trust region filtering during inference: only samples that lie within the trust region (PCH or CGHC) and are predicted positive are classified as positive; all others are classified as negative. Configurations marked with $\ddagger$ retrain the classifier using only samples inside the corresponding trust region and apply the same filtering strategy.

Table 16: Selective classification results across five datasets using CART and MLP classifiers, with and without convex hull-based trust region filtering. The superscript $\dagger$ denotes configurations where the initial classifier, trained on the full dataset, is used in conjunction with trust region filtering during prediction. The superscript $\ddagger$ denotes configurations where the classifier is retrained based on the learned trust region. PCH outperforms CGHC and base classifiers in terms of accuracy, F1 score, and AUC.

| Problem | Index | Trust Region | | CART-based Classification | | | | | MLP-based Classification | | | | |
|---|---|---|---|---|---|---|---|---|---|---|---|---|---|
| | | CGHC | PCH | - | CGHC$^\dagger$ | CGHC$^\ddagger$ | PCH$^\dagger$ | PCH$^\ddagger$ | - | CGHC$^\dagger$ | CGHC$^\ddagger$ | PCH$^\dagger$ | PCH$^\ddagger$ |
| NLNC_N_2 | ACC/% | 91.40 | 91.75 | 98.30 | 98.40 | 98.90 | 98.35 | **99.00** | 94.50 | 94.80 | 97.70 | 95.20 | 98.30 |
| | F1/% | 88.87 | 89.28 | 97.52 | 97.66 | 98.41 | 97.59 | **98.55** | 92.29 | 92.68 | 96.61 | 93.20 | 97.50 |
| | AUC/% | 93.37 | 93.64 | 97.98 | 98.09 | 98.92 | 98.05 | **99.04** | 98.87 | 99.00 | 98.62 | 99.04 | 98.47 |
| Breast Cancer | ACC/% | 95.61 | **97.37** | 91.23 | 92.11 | 96.49 | 91.23 | 95.61 | 94.74 | 92.11 | 91.23 | 94.74 | 93.86 |
| | F1/% | 96.50 | **97.90** | 92.86 | 93.43 | 97.18 | 92.65 | 96.45 | 95.71 | 93.43 | 92.65 | 95.71 | 94.96 |
| | AUC/% | 95.54 | 97.42 | 91.57 | 93.67 | 98.07 | 93.62 | 95.14 | 99.50 | 99.47 | 99.40 | 99.50 | 99.11 |
| Spambase | ACC/% | 81.43 | 89.36 | 91.10 | 92.07 | 91.10 | 93.27 | 91.86 | 93.16 | 92.62 | 90.88 | 93.59 | 89.79 |
| | F1/% | 79.91 | 87.53 | 88.77 | 89.53 | 88.61 | 91.14 | 89.54 | 91.41 | 90.42 | 87.86 | 91.79 | 86.09 |
| | AUC/% | 83.57 | 90.30 | 90.78 | 92.33 | 92.06 | 92.72 | 90.46 | 97.32 | 97.42 | 94.70 | 97.59 | 93.61 |
| Bace | ACC/% | 70.63 | 78.22 | 74.92 | 70.96 | 72.28 | 75.58 | 77.56 | 74.26 | 72.28 | 72.94 | 77.89 | **78.55** |
| | F1/% | 67.87 | **76.43** | 72.26 | 61.40 | 65.00 | 69.92 | 74.05 | 73.10 | 64.41 | 64.35 | 74.72 | 74.71 |
| | AUC/% | 70.42 | 78.16 | 75.78 | 77.23 | 72.07 | 78.12 | 77.85 | 82.87 | 82.46 | 81.14 | 83.37 | 82.44 |
| HIV | ACC/% | - | 95.68 | 95.05 | - | - | **96.89** | 95.81 | 95.76 | - | - | 96.77 | 96.13 |
| | F1/% | - | 39.11 | 32.50 | - | - | 40.19 | 38.72 | 42.12 | - | - | **42.92** | 39.77 |
| | AUC/% | - | 68.59 | 65.59 | - | - | 65.88 | 67.91 | 74.78 | - | - | **75.60** | 74.83 |

**Results and Discussion.** The results from the selective classification experiments across the five datasets are summarized in Table 16. The table shows that the PCH method consistently outperforms CGHC in terms of accuracy (ACC), F1 score, and AUC across most datasets and classifiers. For instance, on the NLNC_N_2 dataset when using CART, the AUC of CART with PCH$^\ddagger$ is 99.04%, compared to 97.98% for only using CART, highlighting a significant improvement when the trust region is utilized.

Moreover, combining the trust region models with the CART classifier generally leads to performance improvements, particularly when retraining the model using only the samples within the trust region (PCH[‡] and CGHC[‡]). This configuration achieves the highest accuracy, F1 score, and AUC across most datasets, clearly demonstrating the advantages of focusing the model's attention on the most relevant data points. However, when comparing the performance of PCH to CGHC, we observe a consistent advantage of PCH across most datasets and classifiers. Specifically, on the Breast Cancer dataset when using MLP,PCH leads to higher classification scores, such as the 78.55% ACC with PCH[‡], compared to the best result of CGHC[‡], which only achieves 72.94% ACC in the same configuration. It is worth noting that when retraining the MLP model, we assign twice the weight to the samples within the trust region during loss calculation, as compared to the samples outside the trust region. This approach helps avoid potential bias in the classification model.

Interestingly, the need for retraining the model is not always critical for achieving strong performance. For datasets with simpler, more linear and convex decision boundaries, such as the Breast Cancer dataset, simply applying the trust region as a classifier is sufficient to achieve excellent results. In this case, PCH significantly outperforms CGHC, as demonstrated by the higher accuracy (97.37% vs. 95.61% for CGHC) and F1 score (97.90% vs. 96.50% for CGHC) achieved by PCH.

In contrast, for datasets with more complex decision boundaries, such as NLNC_N_2, retraining the model on the trust region (PCH[‡]) results in substantial improvements. This configuration leverages the PCH method's ability to tightly encapsulate the relevant data, achieving an AUC of 99.04%, compared to 98.92% AUC for CGHC, further highlighting the superiority of PCH in handling more complex classification tasks.

For higher-dimensional and large-scale datasets, such as HIV, CGHC fails to successfully initialize and compute the trust region. In contrast, PCH remains effective, demonstrating its robustness and scalability in handling challenging, high-dimensional data, where CGHC struggles to produce valid results.

### G.8 CONSTRAINT LEARNING AND EMBEDDING

**Datasets.** We evaluate the integration of trust-region-assisted constraint learning into optimization workflows using the NLNC_N_2 dataset, which approximates the transient stability boundary of a power system. The convex trust regions constructed by PCH and CGHC are those learned in Section G.5 and are visualized in Figure 16. The downstream task is formulated as a two-dimensional linear optimization problem, where the decision variable $(x^{(1)}, x^{(2)}) \in \mathbb{R}^2$ represents key operational states of the power system.

**Experimental Setup.** To simulate uncertainty in real-world power system operations, we randomly sample the upper and lower bounds for each decision variable within predefined ranges. These intervals capture variability in external conditions such as load, generation, or voltage levels. In addition, a random linear cost vector is drawn to reflect fluctuations in economic objectives or operational priorities. The goal is to assess whether constraint learning methods, when combined with different trust-region models, can support effective decision-making under uncertainty. Specifically, we evaluate whether the resulting solutions remain feasible, meaning they lie within the transient stability boundary, and whether they are also cost-effective with respect to the given objective.

The optimization problem is formulated as:

$$\min_{(x^{(1)}, x^{(2)}) \in \mathbb{R}^2} \quad c_1 x^{(1)} + c_2 x^{(2)}$$
$$\text{s.t.} \quad (x^{(1)}, x^{(2)}) \in [\ell, u],$$
$$(x^{(1)}, x^{(2)}) \in \mathcal{R}_{\text{CART/MLP}},$$
$$(x^{(1)}, x^{(2)}) \in \mathcal{C}_{\text{TR}}.$$

where $[\ell, u]$ denotes the uncertainty interval imposed on the decision variables, $\mathcal{R}_{\text{CART/MLP}}$ represents the feasible region specified by the constraint rule learned from either CART or MLP classifiers. The CART and MLP can be modeled as mixed-integer linear constraints in the optimization problem (Jia et al., 2025). $\mathcal{C}_{\text{TR}}$ is the convex trust region obtained from either PCH or CGHC.

For CART, we vary the maximum tree depth from 6 to 10. For MLP, we adopt a two-hidden-layer architecture with hidden layer sizes $(h, 5)$, where $h \in \{15, 30, 50, 100, 200\}$. The MLP is trained using a batch size of 1024, a learning rate of 0.001, and 3000 training epochs.

We compare five configurations for constraint embedding in the optimization model. The baseline configuration, denoted as "–", uses only the constraint learned from the classifier without incorporating any trust region. The configurations marked with $\dagger$ use the classifier trained on the full dataset and apply trust region filtering at the optimization stage, based on either PCH or CGHC. The configurations marked with $\ddagger$ retrain the classifier using only samples within the corresponding trust region, and then embed both the updated classifier constraint and the trust region into the optimization problem. When retraining the MLP within the convex trust region, we apply a sample-weighting strategy that assigns twice the loss weight to samples inside the trust region compared to those outside, thereby encouraging the model to focus on local decision boundaries while retaining global generalization.

Table 17: Performance of constraint learning and embedding with CART and MLP under different configurations. The configuration "–" uses only the learned constraint model without any trust region filtering. The superscript $\dagger$ denotes configurations where the constraint model is trained on the full dataset and combined with trust region filtering during optimization. The superscript $\ddagger$ denotes configurations where the constraint model is retrained based on the corresponding trust region before embedding.

| Index | Depth | CART-based constraint learning | | | | | Hidden size | MLP-based constraint learning | | | | |
|---|---|---|---|---|---|---|---|---|---|---|---|---|
| | | - | CGHC$^\dagger$ | CGHC$^\ddagger$ | PCH$^\dagger$ | PCH$^\ddagger$ | | - | CGHC$^\dagger$ | CGHC$^\ddagger$ | PCH$^\dagger$ | PCH$^\ddagger$ |
| Training Accuracy (%) | 6 | 93.76 | 97.65 | 98.35 | 98.40 | **99.55** | 15 | 99.20 | 99.32 | 98.93 | **99.36** | 99.17 |
| | 7 | 96.46 | 98.03 | 98.80 | 98.74 | **99.75** | 30 | 99.81 | 99.83 | 99.80 | 99.84 | **99.85** |
| | 8 | 97.43 | 98.48 | 99.04 | 99.00 | **99.89** | 50 | 99.84 | 99.87 | 99.80 | **99.92** | 99.89 |
| | 9 | 98.64 | 98.95 | 99.24 | 99.22 | **99.96** | 100 | 99.83 | 99.85 | 99.89 | 99.87 | **99.94** |
| | 10 | 99.30 | 99.50 | 99.46 | 99.69 | **100.00** | 200 | 99.90 | 99.92 | 99.90 | **99.92** | 99.90 |
| Feasibility Rate (%) | 6 | 69.00 | 69.00 | 56.00 | **84.00** | 71.00 | 15 | 81.00 | 81.00 | 85.00 | 89.00 | **90.00** |
| | 7 | 66.00 | 69.00 | 57.00 | **82.00** | 72.00 | 30 | 87.00 | 88.00 | 90.00 | **94.00** | 93.00 |
| | 8 | 68.00 | 70.00 | 56.00 | **82.00** | 70.00 | 50 | 88.00 | 88.00 | 78.00 | **91.00** | 89.00 |
| | 9 | 64.00 | 66.00 | 57.00 | **73.00** | 70.00 | 100 | 86.00 | 86.00 | 77.00 | 91.00 | **92.00** |
| | 10 | 65.00 | 65.00 | 57.00 | **75.00** | 70.00 | 200 | 88.00 | 87.00 | 86.00 | **92.00** | 90.00 |
| Objective Cost | 6 | 5.88 | 5.97 | 5.74 | 6.00 | 5.76 | 15 | 5.78 | 5.78 | 5.78 | 5.78 | 5.78 |
| | 7 | 5.70 | 5.77 | 5.75 | 5.79 | 5.76 | 30 | 5.77 | 5.77 | 5.77 | 5.77 | 5.77 |
| | 8 | 5.72 | 5.76 | 5.73 | 5.78 | 5.76 | 50 | 5.77 | 5.77 | 5.76 | 5.77 | 5.77 |
| | 9 | 5.71 | 5.75 | 5.74 | 5.77 | 5.76 | 100 | 5.77 | 5.77 | 5.76 | 5.77 | 5.77 |
| | 10 | 5.72 | 5.74 | 5.74 | 5.76 | 5.76 | 200 | 5.77 | 5.77 | 5.77 | 5.77 | 5.77 |
| Constraint Number | 6 | 75 | 90 | 90 | 90 | 107 | 15 | 85 | 100 | 100 | 100 | 100 |
| | 7 | 162 | 177 | 158 | 177 | 195 | 30 | 145 | 160 | 160 | 160 | 160 |
| | 8 | 238 | 253 | 290 | 253 | 270 | 50 | 225 | 240 | 240 | 240 | 240 |
| | 9 | 381 | 396 | 409 | 396 | 325 | 100 | 425 | 440 | 440 | 440 | 440 |
| | 10 | 568 | 583 | 463 | 583 | 365 | 200 | 825 | 840 | 840 | 840 | 840 |
| Binary Variable Number | 6 | 12 | 12 | 12 | 12 | 16 | 15 | 20 | 20 | 20 | 20 | 20 |
| | 7 | 24 | 24 | 21 | 24 | 28 | 30 | 35 | 35 | 35 | 35 | 35 |
| | 8 | 33 | 33 | 37 | 33 | 37 | 50 | 55 | 55 | 55 | 55 | 55 |
| | 9 | 48 | 48 | 50 | 48 | 43 | 100 | 105 | 105 | 105 | 105 | 105 |
| | 10 | 66 | 66 | 55 | 66 | 47 | 200 | 205 | 205 | 205 | 205 | 205 |

**Results and Discussion.** Table 17 summarizes the results across key metrics, including training accuracy, feasibility rate of the optimized solutions, average objective cost, and model complexity in terms of the number of constraints and binary variables. The feasibility of each optimized solution is evaluated against the ground-truth stability boundary used to generate the NLNC_N_2 dataset.

The results demonstrate that integrating a trust region into the constraint learning and embedding process significantly improves both safety and efficiency of the downstream optimization. Without any trust region filtering (denoted by "–"), feasible rates remain relatively low, e.g., 69% for CART and 81–88% for MLP. By incorporating the convex trust region learned via PCH ($\dagger$), the feasible rates increase substantially, reaching up to 84% for CART and 94% for MLP. Moreover, this improvement is achieved without retraining the classifier and with a marginal increase in model complexity. This result highlights that even without retraining, PCH-based trust regions can already provide strong regularization for decision boundaries, enabling safe and efficient optimization.

Compared to CGHC, PCH consistently demonstrates superior performance across all tested configurations and classifier types in constraint learning and embedding. In terms of feasibility rate, PCH achieves the highest values under both filtering-only ([†]) and retraining ([‡]) settings. For instance, with CART depth 6, the feasible rate increases from 69.0% (no trust region) to 84.0% with PCH[†], compared to only 69.0% for CGHC[†]. Similar trends hold for MLP classifiers. At hidden size 30, PCH[†] achieves a feasible rate of 94.0%, outperforming CGHC[†] at 90.0%.

Across both CART and MLP configurations, all methods achieve comparable objective cost, with PCH often matching or slightly improving over CGHC. Notably, under deeper tree or wider network settings, PCH tends to reduce the number of binary variables or constraints required to achieve similar performance. For example, at CART depth 10, PCH[‡] achieves the same objective value (5.76) as CGHC[‡], while reducing the number of binary variables from 55 to 47 and constraints from 463 to 365, and also has higher feasibility rate (70% vs. 57%). These results indicate that PCH can produce more compact constraint representations, improving embedding efficiency in downstream optimization without sacrificing feasibility or reliability.

## G.9 ADDITIONAL GRADIENT-BASED BASELINES AND ABLATION STUDY

We provide additional experimental analyses that complement the main results in this subsection. We first compare PCH with the most relevant gradient-based polytope learners available in the literature, highlighting the differences in modeling assumptions and decision boundaries. We then present an ablation study that isolates the contribution of each major component in the PCH framework. Together, these results offer a clearer view of how PCH relates to existing gradient-based approaches and why the full design of the method is necessary for achieving both tightness and scalability.

**Training settings.** All experiments in this subsection use the same polyhedral datasets and the training settings described in Appendix G.3. For comparison with existing gradient-based polytope learners, we include the Convex Polytope Tree (CPT) model of Armandpour et al. (2021), trained using its default settings for 20 epochs. Although CPT is not designed to output explicit linear constraints, it represents the closest gradient-based formulation available in the literature. We report results for three depth configurations (CPT_1, CPT_4, CPT_8), corresponding to increasing levels of model capacity. To further isolate the effect of PCH's design choices, we evaluate three ablated variants: PCH_WOP (removing projection-based weight assignment), PCH_WOS (disabling subregion assignment), and PCH_WOA (removing adaptive structure adjust). All ablated models share the same training settings as the full PCH model. This unified setup ensures that differences in performance can be attributed directly to modeling choices rather than implementation details.

**Main Results.** As shown in Table 18, across all tested polyhedral datasets, PCH achieves the highest trust-region accuracy, demonstrating its ability to learn tight and reliable polyhedral separators. The CPT models improve with increasing tree depth but remain noticeably below the performance of a single PCH hull, reflecting the difficulty of recovering explicit polyhedral boundaries through noisy-OR aggregation. The ablation variants further confirm the importance of PCH's key components: removing projection-based weight assignment (PCH_WOP), disabling subregion assignment (PCH_WOS), or turning off adaptive structure adjustment (PCH_WOA) leads to substantial performance degradation, particularly in higher dimensions. These results underscore that the full PCH pipeline is essential for obtaining accurate and geometrically tight polyhedral trust regions.

Table 18: Trust region accuracy comparison between PCH, CPT models with different tree depths, and ablated variants of PCH on the polyhedral datasets.

| Problem | PCH | CPT_1 | CPT_4 | CPT_8 | PCH_WOP | PCH_WOS | PCH_WOA |
|---------|-----|-------|-------|-------|---------|---------|---------|
| PH_15_2 | **100.00** | 74.70 | 83.67 | 89.00 | 95.19 | 99.41 | 99.98 |
| PH_15_3 | **100.00** | 69.64 | 79.17 | 85.98 | 86.23 | 99.34 | 99.98 |
| PH_15_4 | **100.00** | 66.81 | 73.41 | 80.21 | 82.90 | 99.57 | 99.82 |
| PH_15_5 | **99.52** | 68.19 | 71.94 | 79.04 | 83.06 | 98.96 | 99.11 |
| PH_15_6 | **100.00** | 62.64 | 69.25 | 76.24 | 77.16 | 98.29 | 96.29 |
| PH_15_7 | **99.67** | 66.56 | 69.04 | 75.71 | 77.61 | 94.34 | 97.97 |
| PH_15_8 | **100.00** | 71.81 | 75.01 | 78.69 | 79.56 | 97.01 | 92.15 |
| PH_15_10 | **100.00** | 73.86 | 74.65 | 77.08 | 77.20 | 98.28 | 88.72 |
| PH_15_100 | **99.95** | 64.34 | 65.11 | 65.48 | 71.03 | 87.88 | 55.14 |

## H   USE OF LARGE LANGUAGE MODELS

Large language models (LLMs) were used in this work only as general-purpose assistants to aid and polish the writing. All research ideas, technical developments, theoretical results, and experiments were conceived and conducted by the authors. The authors take full responsibility for the entire content of the paper.

