# OpenReview forum: "Scalable and Adaptive Trust-Region Learning via Projection Convex Hull"
_ICLR.cc/2026/Conference — ICLR 2026 Poster_

### Official Review · Reviewer_G8zX · 2025-10-30

**Soundness:** 3
**Presentation:** 3
**Contribution:** 3
**Rating:** 8
**Confidence:** 3

**Summary:**

This paper proposes the Projection Convex Hull (PCH) method, a scalable framework for learning polyhedral trust regions in high-dimensional spaces. The link between the intractable MINLP for tight convex hull learning and an unconstrained surrogate objective is established. The PCH framework incorporates (i) partition-based subregion assignment, (ii) a surrogate objective to align hyperplanes with class boundaries,and (iii) adaptive hyperplane pruning and addition. Experiments on synthetic and real-world datasets show that
 PCH achieves high accuracy and superior scalability.

**Strengths:**

1. PCH is proposed with sound theoretical foundation.
2. PCH scales to high-dimensional and large-scale problems.
3. Besides convex hull learning, downstream tasks including selective classification and constraint learning are considered.
4. Figure 1 and Figure 2 intuitively illustrates the main steps of PCH .

**Weaknesses:**

1. Whether trapping into local minima is possible during the optimization is not discussed.
2. How is the robustness of PCH in the presence of data noise?

**Questions:**

See Weaknesses.

---

> ### Author Response · Authors · 2025-11-20
> **Response to Reviewer G8zX**
>
> **We sincerely thank the reviewer for recognizing our work, and we address your concerns below.**
>
> **W1. On the possibility of getting trapped in local minima**
>
> We appreciate the reviewer’s concern. As with most nonconvex formulations, the surrogate objective may in principle admit multiple stationary points. In practice, however, we find that the optimization behaves very stably. Each update improves the underlying separation geometry by pushing the hyperplane toward a boundary-supporting direction, and we do not observe the optimization getting stuck in poor local minima in practice.
>
> To illustrate this, we include a 2D toy example in Appendix G.2 that visualizes the optimization trajectory under multiple random initializations, as shown in Figure 7. In all runs, the hyperplanes converge to the same tight supporting configuration, and no unstable behavior is observed. This indicates that the geometric structure of the updates naturally guides the algorithm toward high-quality stationary solutions. Additionally, across all synthetic and real-world datasets evaluated in the paper, the method converges to stable and accurate polyhedral boundaries.
>
> We also adopt two practical strategies to mitigate the risk of local minima further. First, we use a structured initialization scheme based on reliable negative sample selection (Appendix E) to provide a good starting geometry for each hyperplane. Second, multiple restarts may be used when needed.
>
> **W2. On the robustness of PCH in the presence of data noise**
>
> We thank the reviewer for raising this critical point. Structurally, PCH is naturally robust to moderate noise. The surrogate objective emphasizes boundary-supporting samples, and the local subregion assignment reduces the influence of outliers far from the active hyperplane. As a result, moderate noise in the negative class has a limited effect on the learned supporting hyperplanes. This behavior is also aligned with the geometric interpretation of PCH, where the hyperplanes are encouraged to follow the tight outer envelope of the positive class.
>
> We evaluate PCH on polyhedral datasets with injected noise and observe that the optimization remains stable and the trust-region accuracy continues to improve throughout training. As shown in Table 8, across dimensions 2-100, PCH maintains high accuracy under noise, whereas several baselines show noticeable degradation.
>
> In addition to synthetic tests, our experiments on Breast Cancer, Spambase, Bace, HIV, and additional real-world datasets such as MNIST-35, FMNIST-24, CIFAR-35, and Adult Income also demonstrate strong robustness. These datasets may naturally contain feature irregularities and moderate label imperfections, yet PCH consistently attains the highest trust-region accuracy.
>
> We have included the detailed noisy experiments and the corresponding visualizations in Appendix G.3 of the revised manuscript. The combined synthetic and real-data evidence indicates that PCH remains robust under substantial data noise.
>
> **We thank the reviewer again for the helpful comments, which guided us to clarify the optimization behavior and the robustness properties of PCH in the revised manuscript.**

---

### Official Review · Reviewer_5jY7 · 2025-10-31

**Soundness:** 3
**Presentation:** 3
**Contribution:** 3
**Rating:** 6
**Confidence:** 2

**Summary:**

The paper proposes an algorithm for learning convex hulls (polyhedral trust regions) from labeled data, ensuring that all positive samples are enclosed while excluding negatives as much as possible. Traditional geometric or MILP approaches fail to scale in high dimensions and lack flexibility, and it aims to design a scalable, theoretically grounded, and adaptive framework to learn boundary-tight convex hulls efficiently. The main contribution is that, the authors propose a divide-and-conquer learning framework that links an exact MINLP problem to an unconstrained surrogate objective, making gradient-based optimization feasible.

**Strengths:**

The paper presents a novel gradient-based surrogate formulation for convex hull learning, bridging discrete MINLPs with differentiable optimization. Moreover, its adaptive structure and surrogate update scheme enable practical training in high-dimensional settings, as demonstrated empirically.

**Weaknesses:**

1. It is unclear how the main theoretical result (Theorem 1) relates to the practical algorithm. The theorem establishes the existence of a certain weight assignment, but it is not evident whether this weight can be computed or approximated by Algorithm 1 or the procedure described in Section 5.2. Although there is some discussion in lines 354–356, the connection remains ambiguous, especially since Equation 3 does not explicitly include a weight term.

2. In addition, no convergence analysis or guarantee (e.g., monotonic improvement) is provided for the surrogate updates.



3. Since the algorithm is somewhat unintuitive, it would be helpful to include a simple 2D toy example illustrating the evolution of the hyperplane and weight updates over iterations. This visualization would greatly enhance readers’ understanding of the surrogate optimization process.

**Questions:**

1. A main contribution concerns the algorithm’s computational complexity. However, it appears to depend on the number of iterations—what is the typical order or empirical scaling of this term?

2. The authors should discuss potential failure cases, such as when the positive region is nonconvex or when the hyperplane budget is insufficient to capture the desired boundary.

3. Is that any small example showing that the surrogate formulation indeed approxiamte the discrete MINLP problem?

---

> ### Author Response · Authors · 2025-11-20
> **Response to Reviewer 5jY7 (Part I)**
>
> **We thank the reviewer for the insightful suggestions and address the concerns below.**
>
> **W1. On the connection between Theorem 1 and the practical algorithm**
>
> We thank the reviewer for raising this point. Theorem 1 provides a one-directional theoretical statement (the MINLP solution ⇒ weight assignment). This result establishes a structural relation: any optimal hyperplane of the MINLP must satisfy a stationary condition of the form $C\omega = d$ (Eq. (5)) and  $a \approx \sum_i \omega_i\, p_i^L p_i^R\, y_i x_i$ (Eq. (6)), which links the hyperplane parameters with a weight vector.
>
> This stationary condition is what connects the theory to the algorithm. Eq. (6) allows us to express the separation gap as a function of the weights, so that maximizing the gap can be carried out in the weight space rather than by solving the constrained NLP in Eq. (3) over $(a,b)$. Algorithm 1 uses this relation to perform projected gradient ascent on the weights, followed by gradient descent on $(a,b)$. In this sense, the algorithm approximates the ideal weight assignment whose existence is guaranteed by Theorem 1.
>
> Regarding Eq. (3), while the weight does not explicitly appear, the formulation can be interpreted as a nearest-point problem (NPP), where the associated dual variables share a similar geometric meaning as the weights. Because Eq. (3) is challenging to warm-start or solve repeatedly, we employ its unconstrained surrogate form in Eq. (4), and Lemma 3 justifies the equivalence between Eq. (3) and the surrogate together with the stationary conditions in Eq. (5)–(6). Algorithm 1, therefore, provides a tractable implementation of this surrogate–dual relation.
>
> We have revised Sections 4 and 5 to make this connection more explicit.
>
> **W2. On the convergence analysis of the surrogate updates**
>
> We agree that a brief analysis of convergence will improve clarity. Our algorithm alternates between updating the hyperplane parameters through a gradient step on the surrogate objective and updating the weights by maximizing the separation gap. Although these two updates originate from different formulations, both steps consistently push hyperplanes toward a boundary-supporting direction with a larger margin. As a result, the overall procedure behaves stably and consistently improves the same separation geometry across iterations.
>
> For the hyperplane update, the surrogate objective is smooth (under a fixed subregion partition) and lower-bounded, and a sufficiently small learning rate ensures that each step decreases its value. The weight update increases the separation gap, which enhances the same geometric objective that the surrogate seeks to improve. This alternating improvement is in line with the classical perspective of block-wise optimization [1], where each update improves a local criterion and the iterates tend to stabilize as both the surrogate and separation gap stop changing appreciably.
>
> To complement this theoretical intuition, we follow the concern in **W3** and provide empirical evidence by expanding the 2D visualization in Figure 2 to visualize the evolution of the hyperplanes and weight assignments across iterations. Both sets of trajectories rapidly settle into stable shapes. These intermediate visualizations provide clear empirical evidence that the alternating updates converge to a steady configuration in practice.
>
> We have included this convergence discussion in Section 5, and the corresponding empirical visualization and illustration have been added in Appendix G.2 in the revised manuscript.
>
> [1] Razaviyayn, M., Hong, M., and Luo, Z.-Q. “A unified convergence analysis of block successive minimization methods for nonsmooth optimization.” SIAM J. Optim., 2013.
>
> **W3. On the need for a simple 2D toy example**
>
> We appreciate the reviewer’s suggestion. The proposed method is conceptually simple in spirit, as each iteration refines the supporting hyperplane and updates the corresponding weights in a structured manner. Even so, we agree that a visual illustration can further clarify how the optimization proceeds. In the revised version, we include a 2D toy example in Appendix G.2 that displays the evolution of both the hyperplanes and the weight assignment profiles across iterations.
>
> The intermediate snapshots show how each hyperplane progressively aligns with the actual boundary and how the weight assignment profiles stabilize as the surrogate converges. This example provides a clear, step-by-step visualization of how the alternating updates cooperate to improve the polyhedral approximation. We believe this addition substantially enhances the interpretability of the surrogate optimization process.

---

> ### Author Response · Authors · 2025-11-20
> **Response to Reviewer 5jY7 (Part II)**
>
> **Q1. On the computational complexity and the number of iterations**
>
> We thank the reviewer for highlighting the importance of iteration complexity. In practice, the number of iterations required is small. Across all high-dimensional experiments, we run 15 outer rounds, each containing 100 inner gradient updates. We observe that the algorithm typically stabilizes within the first 3–5 rounds. The remaining rounds are included only for robustness and fair comparison. We include training dynamics against iteration in Appendix G. As shown in Figures 9 and 12, the trust-region accuracy converges rapidly. The algorithm scales well with dimension.
>
> **Q2. On potential failure cases**
>
> We appreciate the reviewer’s suggestion to discuss possible failure cases. When the positive region is nonconvex, PCH does not fail; it naturally returns the convex outer envelope of the region. This behavior is inherent because we want full positive inclusion, which must necessarily include negatives lying inside $\mathrm{conv}(\mathcal{D}\_+)$. PCH is explicitly designed to produce such a convex outer approximation: it guarantees full positive coverage while maximizing separation from external negatives $\mathcal{D}\_-^{\mathrm{out}}$. In the 2D examples included in Appendix G.5, PCH closely tracks the outer boundary even when the ground truth is nonconvex, and we do not observe noticeable degradation in our experiments.
>
> A genuine limitation arises when the hyperplane budget $S$ is insufficient to represent the desired boundary. In this case, the learned polyhedron becomes a coarse approximation and may include additional negatives. This phenomenon is expected for any polyhedral method with limited capacity. To illustrate this, we increase the hyperplane budget $S$ from 4 to 15 to fit a 2D circular boundary in the revised manuscript. The target boundary is smooth and cannot be represented exactly with a small number of hyperplanes. As $S$ increases from 4 to 15, the trust-region accuracy improves monotonically (from 87.6% to nearly 99.4%), and the polyhedron visually approaches the circular boundary. These visualizations are provided in Appendix G.4.
>
> **Q3. On whether the surrogate approximates the discrete MINLP solution**
>
> We thank the reviewer for this insightful question. Following your concern in **W3**, we construct a simple 2D toy example in Appendix G.2. Since solving the exact MINLP directly is difficult even in two dimensions, we instead solve a relaxed MILP formulation that minimizes the number of included negatives. This relaxation provides an approximate reference boundary for the discrete MINLP. We then train PCH on the same dataset using three hyperplanes.
>
> As shown in the visual comparison in Figure 8, the PCH boundary tightly supports the positive samples and passes almost exactly through the positive support vectors. In contrast, the MILP-relaxed boundary is noticeably looser and maintains a visible margin away from the supporting positive points, indicating a lack of geometric tightness.
>
> These observations provide empirical evidence that the surrogate formulation effectively approximates the discrete MINLP at the level of its supporting-hyperplane geometry.
>
> **We thank the reviewer again for the detailed and thoughtful questions, which helped us clarify the theoretical role of the surrogate, its convergence behavior, and its practical limitations in the revised manuscript.**

---

### Official Review · Reviewer_c9H4 · 2025-11-01

**Soundness:** 3
**Presentation:** 2
**Contribution:** 3
**Rating:** 4
**Confidence:** 2

**Summary:**

Motivated by the observation that complex and nonlinear classification boundaries often exhibit regional linearity and can be locally approximated by hyperplanes with small classification error, the paper constructs a convex hull that is scalable,
structurally adaptive, and geometrically tight. Specifically, from the MINLP problem formulation, the work establishes its connection to a family of unconstrained surrogate objectives. This enables gradient-based learning of convex hulls that are both compact and theoretically motivated.

**Strengths:**

The paper effectively addresses key limitations of prior works in terms of scalability, adaptivity, and boundary tightness, and provides theoretical guarantees to support its claims. Given the intrinsic complexity of the MINLP formulation, it is particularly commendable that the authors decompose the original problem into a series of per-hyperplane subproblems with unconstrained surrogate objectives, making the optimization process more tractable.
The final solution is obtained through gradient-based optimization, which further enhances computational efficiency. Moreover, the proposed adaptive structural adjustment introduces principled hyperplane addition and pruning criteria, enabling the model to maintain scalability without sacrificing geometric tightness.

**Weaknesses:**

The experimental evaluation appears somewhat limited. While the results on the BreastCancer, Spambase, Bace, and HIV datasets demonstrate the feasibility of the proposed approach, these datasets are relatively simple and well-studied binary classification benchmarks. To better assess the generalization ability and robustness of the proposed method, it would be valuable to include experiments on more challenging and diverse datasets, such as those with higher-dimensional features, class imbalance, or more complex data manifolds.

Additionally, it is not entirely clear whether the optimization process is guaranteed to converge. Since the proposed formulation involves nonconvex components and relies on surrogate objectives and iterative weight updates, a brief theoretical or empirical discussion of the convergence behavior (e.g., monotonicity of the objective, stability of fixed points, or stopping criteria) would strengthen the work and improve the reader’s understanding of the algorithm’s reliability.

**Questions:**

Please see Weaknesses.

---

> ### Author Response · Authors · 2025-11-20
> **Response to Reviewer c9H4**
>
> **We thank the reviewer for the thoughtful comments and address the concerns below.**
>
> **W1. On more challenging and diverse experimental datasets**
>
> We appreciate the reviewer’s suggestion to evaluate the proposed method on more challenging and diverse datasets. To further illustrate the scalability of our approach, we conducted additional experiments on several high-dimensional datasets with complex feature distributions, including MNIST, Fashion-MNIST (FMNIST), CIFAR10, and Adult Income.
>
> We convert MNIST, FMNIST, and CIFAR10 into binary classification tasks by selecting label pairs (3,5) for MNIST-35, (2,4) for FMNIST-24, and (3,5) for CIFAR-35. MNIST-35 and FMNIST-24 use raw pixel features (dimension 784), whereas CIFAR-35 uses three-channel image features (dimension 3072). The Adult Income dataset is widely used for evaluating model performance under strong class imbalance. These tasks collectively cover high dimensionality, complex data distributions, and substantial class imbalance.
>
> Classical polyhedral baselines (QH, CGHC, DCH, PinCHD) do not scale to these high-dimensional tasks within a reasonable time budget. Therefore, we compare against **DeepHull**, the only baseline that scales to these settings. Results are summarized below (with full details included in Appendix G.6):
>
> | Problem      | MNIST-35 | FMNIST-24 | CIFAR-35 | Adult Income |
> |--------------|----------|-----------|----------|--------------|
> | **PCH**      | **99.98**| **99.10** | **92.58**| **89.08**    |
> | **DeepHull** | 99.84    | 91.84     | 68.66    | 64.66        |
>
> Although DeepHull is not designed to produce explicit polyhedral constraints or guarantee full positive inclusion, we include it as the only scalable gradient-based baseline available for these datasets.
>
> These results confirm that the proposed method achieves superior trust-region accuracy and hull tightness on all newly added tasks, while still providing an explicit polyhedral trust region. These results demonstrate that PCH scales effectively to high-dimensional, complex, and imbalanced settings while maintaining explicit polyhedral structure.
>
> **W2. On the convergence behavior of the optimization process**
>
> We thank the reviewer for raising the question on convergence. Our algorithm alternates between updating the hyperplane parameters via a gradient step on the surrogate objective and updating the weights by maximizing the separation gap. Although these two updates originate from different formulations, both updates consistently improve the same underlying separation geometry by pushing each hyperplane toward a boundary-supporting direction with a larger margin. The overall optimization, therefore, behaves stably across iterations.
>
> For the hyperplane update, the surrogate objective is smooth (under a fixed subregion partition) and lower-bounded, and a sufficiently small learning rate ensures that each step decreases the surrogate value. When the surrogate stabilizes, it approximately satisfies the stationary condition in Eq. (6), which relates the hyperplane parameters to the current weights. This relationship allows us to express the separation gap as a function of the weights, upon which we apply gradient ascent to increase the gap. Thus, while the two updates do not optimize a single unified analytic objective, both contribute to a consistent geometric improvement. Under this view, the fixed-structure part of the procedure aligns with the classical alternating optimization framework [1], which guarantees convergence to a first-order stationary configuration under sufficiently small step sizes.
>
> To complement the theoretical justification, we have expanded the 2D visualization in Figure 2 to show the empirical convergence of both hyperplane parameters and weight assignments. In this example, we visualize the trajectories of the hyperplanes and weight assignments over iterations. Both the hyperplanes and the weight profiles progressively stabilize and eventually stop changing, indicating that the procedure reaches a steady configuration. This provides explicit empirical confirmation that the optimization process converges in practice.
>
> We have included this convergence discussion in Section 5, and the corresponding empirical visualization and illustration have been added in Appendix G.2 in the revised manuscript.
>
> [1] Razaviyayn, M., et al. "A unified convergence analysis of block successive minimization methods for nonsmooth optimization". SIAM Journal on Optimization 2013.
>
> **We thank the reviewer again for the constructive comments, which helped us strengthen both the experimental evaluation and the discussion of convergence in the revised manuscript.**

---

> ### Comment · Reviewer_c9H4 · 2025-11-26
>
> Thank you very much for your thoughtful and detailed response. I would like to ask a follow-up question.
>
> At lines 350–354 and in Appendix D, the connection between Theorem 1 and the proposed algorithm is explained. However, it still remains unclear to me whether the $w$ that maximizes the separation gap is indeed the same $w$ that ensures the stationary point of (4) becomes the optimal solution of (3). I have also read the author's responses to similar questions raised by other reviewers, but I am still uncertain whether I fully understand the precise relationship. I may very well be overlooking an important point; therefore, I would greatly appreciate it if you could offer a more rigorous, formula-based clarification of this connection.
>
> Additionally, if time permits, could you also share experimental results for an outlier detection task in addition to the classification setting?
>
> Thank you again for your time and assistance.

---

> > ### Author Response · Authors · 2025-11-28
> > **Response to Reviewer c9H4**
> >
> > **We sincerely thank the reviewer for the careful reading of our manuscript and for the follow-up questions. Below we address the two concerns in detail.**
> >
> >
> > **On the role of $\omega$ and weight assignment**
> >
> > We summarize the main clarification of the connection as follows.
> > In the idealized formulation described in Appendix D, for each weight vector $\omega$ we solve the surrogate objective (4) to stationarity and obtain a hyperplane $(a(\omega),b(\omega))$. The separation gap can then be written as a function of $\omega$ via $\xi(\omega) := \xi(a(\omega),b(\omega))$. Maximizing this gap over feasible $\omega$ recovers a hyperplane that coincides with the optimal solution of the constrained separation problem (3). Our practical algorithm is a computationally efficient approximation of this ideal two-level scheme.
> >
> > Below we provide a more rigorous, formula-based clarification of this connection.
> >
> > In the constrained separation problem (3), we seek a hyperplane $(a^\star, b^\star)$ that maximizes the geometric separation gap:
> > $$
> > \xi(a,b) := \min_{x\in\mathcal D^+} a^\top x - \max_{x\in\mathcal D_s^-} a^\top x,
> > \qquad \|\|a\|=1.
> > $$
> > For a fixed weight vector $\omega$, the surrogate objective $f(a,b;\omega)$ in (4) admits stationary points characterized by the linear constraints $C\omega=d$ in (5) and the fixed-point relation in (6):
> > $$
> > a = g(a,\omega) := \sum_i \omega_i\, p_i^{\text{L}} p_i^{\text{R}} y_i x_i,
> > $$
> > as established in Lemma 2 and Lemma 3. Under these conditions, $(a,b)$ is a stationary point of the surrogate for the given $\omega$, and we can denote such a stationary point as $(a(\omega),b(\omega))$. Hence, the separation gap can be viewed as a function of $\omega$:
> > $$
> > \xi(\omega) := \xi(a(\omega),b(\omega)).
> > $$
> > As outlined in Appendix D, this yields an ideal two-level optimization scheme:
> > $$
> > \max_{\omega: C\omega=d}\ \xi(a(\omega), b(\omega)) \\
> > \text{s.t.}\quad (a(\omega), b(\omega)) \in \arg\min_{a,b} f(a,b;\omega).
> > $$
> >
> > On the lower level, for any $\omega$ satisfying $C\omega=d$, the surrogate (4) is (ideally) solved to stationarity, so that the fixed-point relation (6) holds. This allows $a(\omega)$ and $b(\omega)$ to be regarded as implicit functions of $\omega$. On the higher level, we apply gradient ascent on $\xi(\omega)$:
> > $$
> > \omega \leftarrow  \omega + \eta\\,\frac{\partial\xi(\omega)}{\partial\omega},
> > $$
> > followed (in its simplest form) by projection onto the affine subspace defined by $C\omega=d$:
> > $$
> > P_S(\omega)=\omega-C^{\top}\left(C C^{\top}\right)^{-1}(C \omega-d).
> > $$
> >
> > Our practical algorithm is a computationally efficient approximation of this two-level scheme. Instead of fully solving $\nabla_{a,b} f(a,b;\omega)=0$ for each $\omega$, we take a single gradient step on the surrogate to move $(a,b)$ toward stationarity, and then update $\omega$ by projected gradient ascent on $\xi(\omega)$ using the chain rule based on (6). Although this alternating procedure does not guarantee that the final $\omega$ is exactly equal to the theoretical maximizer, it is specifically designed so that the learned $\omega$ approximates the weight assignments whose existence is guaranteed by Theorem 1, i.e., those that make the stationary point of (4) coincide with an optimal solution of (3).
> >
> > **Consequently, the resulting surrogate hyperplanes align with the optimal separation geometry encoded in (3).**
> >
> >
> > **On outlier detection**
> >
> > We would like to clarify that our method is designed for *supervised* separation problems, whereas outlier detection is typically formulated as an *unsupervised* or *one-class* task with different modeling assumptions. Applying our framework to such settings would require substantial reformulation, so unsupervised outlier detection is outside the intended scope of our method.
> >
> > Nevertheless, since some anomaly datasets provide explicit labels and can be treated as supervised binary classification, we additionally report trust-region performance on two such benchmarks (UCI Arrhythmia and Credit Card Fraud). The results (accuracy, with the number of learned hyperplanes in parentheses) are:
> >
> > | Dataset        | CGHC/%     | PinCHD/%     | DeepHull/%      | PCH/%        |
> > |----------------|:------------:|:--------------:|:-------------:|:--------------:|
> > | Arrhythmia     | 94.47 (3)    | 92.26 (149)    | 100.00 (3)    | **100.00 (3)** |
> > | CreditCard     | –            | 99.34 (149)    | 77.81 (15)    | **99.88 (15)** |
> >
> > Even though anomaly detection is not our target domain, the proposed PCH achieves strong separation performance on these labeled anomaly-classification tasks, while requiring fewer hyperplanes.
> >
> >
> > **We sincerely appreciate the reviewer’s questions, which allowed us to improve the clarity of both the theoretical connection and the applicability of our framework.**

---

### Official Review · Reviewer_hxmb · 2025-11-01

**Soundness:** 3
**Presentation:** 3
**Contribution:** 3
**Rating:** 8
**Confidence:** 3

**Summary:**

This paper introduces the Projection Convex Hull (PCH), a scalable and adaptive framework for learning compact, polyhedral trust regions from data consisting of positive and negative points. The fundamental problem of computing the tight convex hull is actually a Mixed-Integer Nonlinear Program (MINLP). Classical geometric methods suffer from complexity that grows exponentially with dimension, and existing optimization-based approaches lack practical scalability.

The authors overcome this challenge through a rigorous theoretical decomposition. They first formally express the convex hull problem as a MINLP that explicitly accounts for model compactness. Then, they reduce the problem into multiple subproblems that handle one hyperplane at once. The authors prove that the solution for finding the optimal hyperplanes can be recovered as a stationary point of an unconstrained surrogate objective function under suitable weights. This surrogate loss function is minimized using gradient steps. The weight for each negative point is carefully designed and updated during this process to ensure that minimizing the weighted surrogate function is equivalent to maximizing the intended separation margin of the original MINLP.

In practice, the PCH framework achieves structural adaptivity: it dynamically manages the model complexity by pruning redundant hyperplanes and adding new ones if the current hull still incorrectly encloses negative samples. This ensures the final model is as compact and accurate as possible.

**Strengths:**

- The paper is very well written. It solves the problem gradually by breaking it down into simpler, smaller subproblems, makes the algorithm easy to follow.
- The method is strong in both theory and practice. PCH maintained high accuracy and low running time and its usage benefits the real-world downstream tasks.

**Weaknesses:**

- More detailed comparison to related gradient-based works are needed.
- see the question below

**Questions:**

- Lemma 1 does not ensure that the collection of all optimal solutions of the subproblem is the optimal solution of the original problem. How does the proposed algorithm ensure its global optimality?
- The adaptive strategy adds hyperplane when contains negative points and removes when encounter redundancy. Then how to decide the number S in practice? And what happened when the positive points are not convex (where the convex hull containing negative points are inevitable) ?

---

> ### Author Response · Authors · 2025-11-20
> **Response to Reviewer hxmb (Part I)**
>
> **We thank the reviewer for the careful reading and the constructive comments. Below, we address the three concerns in turn and clarify how we revise the manuscript.**
>
> **W1. On the need for comparison to gradient-based related works**
>
> We agree that including comparisons to gradient-based baselines would strengthen the evaluation. Explicit polyhedral trust-region learning remains relatively underexplored. Most existing gradient-based convex-region models, such as DeepHull or neural convex approximators, learn implicit convex sets and do not output explicit linear constraints $Ax \ge b$ nor guarantee full inclusion of all positive samples. Therefore, they are not directly comparable to our problem setting, although we still included DeepHull in the original submission for completeness.
>
> To the best of our knowledge, the earliest related gradient-based works are the convex polytope machine (CPM) [1], and its recent extension is convex polytope trees (CPT) [2]. However, even restricting CPT to a single node does not yield a polytope representable as $Ax \ge b$. This is because the noisy-OR aggregation used in CPT yields a log-sum-exp decision boundary, resulting in a smooth, nonlinear boundary rather than a polyhedral one and, hence, cannot be expressed as a finite polyhedral constraint set. Thus, CPT is not a direct substitute for explicit polyhedral trust-region learning, but it provides a gradient-based reference point.
>
> To address the reviewer’s suggestion, we added two new comparisons on the polyhedral separation datasets:
>
> 1. Comparison with CPT at different depths (1, 4, and 8).
>    Although CPT is designed for decision-tree induction rather than trust-region modeling, it remains the closest gradient-based model that attempts to learn convex polytopes.
>    Results show that CPT requires significantly deeper structures to achieve the accuracy of a single PCH hull.
>
> 2. Ablation-based gradient variants of PCH, including: PCH_WOP: without projection-based weight assignment, PCH_WOS: without subregion assignment, PCH_WOA: without adaptive structure adjust.
>    All these variants exhibit degradation in accuracy, demonstrating that PCH’s projection-based weight assignment, subregion-wise optimization, and adaptive structure compression are essential beyond simply applying gradient descent.
>
> The results are summarized below:
>
> | problem    | PCH    | CPT_1 | CPT_4 | CPT_8 | PCH_WOP | PCH_WOS | PCH_WOA |
> |------------|--------|-------|-------|-------|---------|---------|---------|
> | PH_15_2    | **100.00** | 74.70  | 83.67 | 89.00 | 95.19   | 99.41   | 99.98   |
> | PH_15_3    | **100.00** | 69.64 | 79.17 | 85.98 | 86.23   | 99.34   | 99.98   |
> | PH_15_4    | **100.00** | 66.81 | 73.41 | 80.21 | 82.90   | 99.57   | 99.82   |
> | PH_15_5    | **99.52**  | 68.19 | 71.94 | 79.04 | 83.06   | 98.96   | 99.11   |
> | PH_15_6    | **100.00** | 62.64 | 69.25 | 76.24 | 77.16   | 98.29   | 96.29   |
> | PH_15_7    | **99.67**  | 66.56 | 69.04 | 75.71 | 77.61   | 94.34   | 97.97   |
> | PH_15_8    | **100.00** | 71.81 | 75.01 | 78.69 | 79.56   | 97.01   | 92.15   |
> | PH_15_10   | **100.00** | 73.86 | 74.65 | 77.08 | 77.20   | 98.28   | 88.72   |
> | PH_15_100  | **99.95**  | 64.34 | 65.11 | 65.48 | 71.03   | 87.88   | 55.14   |
>
> We have included these comparisons in Appendix G.9 of the revised submission and added a more detailed discussion of gradient-based polytope-learning approaches in Section 2.
>
> [1] Kantchelian, A., et al. “Large-margin convex polytope machine.” NeurIPS 2014.
> [2] Armandpour, M., et al. “Convex polytope trees.” NeurIPS 2021.

---

> ### Author Response · Authors · 2025-11-20
> **Response to Reviewer hxmb (Part II)**
>
> **Q1. On method global optimality**
>
> We appreciate the reviewer’s insightful question. Lemma 1 and Theorem 1 establish a necessary condition:
> any global solution of the MINLP must decompose into hyperplane-wise stationary subproblems.
> This one-directional implication (global ⇒ local) provides an essential theoretical guideline: it reveals the algebraic form that optimal supporting hyperplanes must satisfy, and motivates the surrogate updates used in Algorithm 1. In particular, the weight–hyperplane relationship derived in Eq. (5)–(6) identifies the geometric quantities that must align at optimality, enabling us to replace the combinatorial MINLP with a continuous surrogate that promotes the same structural configuration.
>
> The algorithm therefore does not rely on the converse implication (local ⇒ global). Because the MINLP is computationally intractable in high dimensions, an exact global guarantee is impractical; instead, we aim to design a surrogate whose stationary points mimic the structure of the true optimum, whenever it exists. The projection-based weight update and the surrogate gradient descent provide precisely this alignment mechanism.
>
> Empirically, this theoretical–algorithmic connection proves effective. Across all polyhedral separation tasks, the learned hyperplanes converge to tight supporting directions that coincide with the actual boundary, achieving near-exact separation (>99–100%). This consistent behavior demonstrates that the surrogate converges to high-quality solutions that satisfy the same structural characteristics identified in the MINLP analysis.
>
> We have added more discussion in Section 4 to make this theoretical role explicit.
>
> **Q2. On choosing the hyperplane budget $S$ and handling nonconvex positive regions**
>
> **Choice of \$S$.**
> The number of hyperplanes $S$ serves as a capacity parameter that balances hull tightness with the complexity of the resulting model.
> In practice, we choose $S$ via validation by monitoring trust-region accuracy and selecting the smallest $S$ that achieves the desired performance. This procedure is simple to implement and works consistently across datasets.
> For downstream optimization (e.g., constraint learning), $S$ is additionally chosen based on the allowable number of constraints in the embedded optimization model.
>
> **Nonconvex positive regions.**
> When $\mathcal{D}\_+ $ is nonconvex, any convex outer approximation necessarily includes interior negatives.
> PCH is designed to produce a convex outer approximation: it guarantees full positive coverage while maximizing separation from external negatives $\mathcal{D}\_-^{\mathrm{out}} = \mathcal{D}_- \setminus \mathrm{conv}(\mathcal{D}\_+)$.
> Importantly, PCH operates directly on $\mathcal{D}\_+ \cup \mathcal{D}\_-$ and does not require explicitly computing $\mathrm{conv}(\mathcal{D}\_+)$.
> This behavior is illustrated in our 2D examples (Figure 16 in Appendix G.5), where PCH closely tracks the outer envelope even when the ground truth is nonconvex.
>
> We have added a visualization case study for the 2D nonlinear circular dataset to evaluate the effect of the hyperplane budget in Appendix G.4, and we have discussed the natural applicability of the proposed PCH to nonlinear nonconvex separation tasks in Appendix G.5.
>
> **We thank the reviewer again for the insightful comments, which helped us refine both the theoretical exposition and the empirical evaluation in the revised manuscript.**

---

> > ### Comment · Reviewer_hxmb · 2025-11-28
> >
> > Thank you for the clarification. The additional experiments have resolved my concerns. And my earlier confusion about the theory and details has been cleared up, so I will keep my positive scores.

---

> > > ### Author Response · Authors · 2025-11-28
> > >
> > > Thank you for revisiting our clarifications and experiments. We truly appreciate your positive and constructive feedback.

---

### Author Response · Authors · 2025-11-20
**Response Summary**

We thank all reviewers for their careful evaluations and constructive suggestions. We are encouraged that the reviews consistently recognize the strengths of our submission. Reviewers describe the paper as “very well written” (hxmb), “strong in both theory and practice” (hxmb), “novel” with a “rigorous theoretical decomposition” (hxmb), and “a gradient-based surrogate formulation that bridges discrete MINLPs with differentiable optimization” (5jY7). They highlight the method’s “scalability, adaptivity, and boundary tightness” (c9H4) and note that PCH enables “practical training in high-dimensional settings” (5jY7). Reviewer G8zX also emphasizes the “sound theoretical foundation” and the ability of PCH to “achieve high accuracy on synthetic and real-world datasets” (G8zX), reinforcing both the theoretical contributions and empirical strength of the framework.

Building on these encouraging assessments, we summarize below the main concerns raised by the reviewers and the clarifications and improvements made in the rebuttal and revised manuscript.

**1. Relation between theory and algorithm.**
Several reviewers asked how the theoretical results connect to the practical procedure. We clarified that Theorem 1 formalizes the stationary condition linking the MINLP structure to a feasible weight assignment, which directly motivates the projected weight update in Algorithm 1. Sections 4–5 have been revised to make this connection explicit and to show how the surrogate optimization approximates the stationary configuration suggested by the theory.

**2. Convergence behavior and local optima.**
Reviewers also asked about convergence and the possibility of local optima. We provided a theoretical explanation showing that the alternating updates consistently improve the same separation geometry. Combined with empirical trajectories, where both the hyperplanes and weight profiles stabilize quickly across random initializations, this indicates that the optimization behaves reliably in practice.

**3. Diverse and challenging experiments.**
To address requests for more challenging datasets, we added results on MNIST-35, FMNIST-24, CIFAR-35, and Adult Income, which involve high dimensionality, nonlinear, and imbalanced data manifolds. PCH consistently achieves the highest trust-region accuracy and boundary tightness among scalable baselines. We also included noisy polyhedral experiments, in which PCH remains stable and achieves high separation accuracy. The iteration curves show that PCH typically converges within a few rounds, whereas classical geometric and MILP-based approaches become impractical at these scales.

**4. Algorithm update process visualization.**
Following the request for a clearer illustration of the update process, we added a 2D example showing how the hyperplanes and their corresponding weight assignments evolve during training. This provides a step-by-step visualization of how the surrogate optimization proceeds.

**5. Model capacity and boundary approximation**
We clarified how the hyperplane budget affects boundary quality and added a 2D circular example to illustrate capacity scaling. We also explained why nonconvex positive regions do not pose issues: any trust-region-learning method necessarily returns an outer approximation and includes interior negatives, and PCH follows this established principle while maximizing separation from external negatives.

We thank all reviewers again for their insightful feedback, which helped us improve the paper's presentation, theoretical clarity, and experimental coverage.

**Author–Reviewer Discussion Summary**

Reviewer hxmb expressed that the clarifications and additional experiments fully addressed the theoretical concerns and confirmed maintaining positive scores. Reviewer c9H4 raised a follow-up question, and we provided a detailed, formula-based clarification together with additional results. Although no further discussion was allowed afterward, the discussion helped clarify key theoretical aspects of the method and provided a direct response to the points raised in the follow-up question.

---

### Meta-Review · Area_Chair_5aCv · 2026-01-06

**Summary:**

This paper proposes Projection Convex Hull (PCH), a scalable and adaptive framework for learning polyhedral trust regions from labeled data. By connecting an exact but intractable MINLP formulation to a differentiable surrogate objective, the method enables gradient-based learning of compact, boundary-tight convex hulls with strong theoretical motivation and practical scalability.

The reviewers agree that the paper is technically strong, clearly written, and addresses an important problem. They highlight the principled theoretical development, the divide-and-conquer optimization strategy, and the convincing experimental results demonstrating scalability and compactness. Concerns mainly centered on clarity of the theory–algorithm connection, comparisons to gradient-based baselines, and robustness.

The authors’ responses carefully addressed these points with clarifications, added experiments, and improved exposition, resolving the reviewers’ concerns. Based on the consistently positive reviews, strengthened revision, and clear contribution to trust-region learning, I am pleased to recommend acceptance.

**Reviewer Concerns:**

Please see my summary.

**Reviewer Scores:**

It is difficult to say.  Overall, the authors provided some solid rebuttal, but it's a subjective judgement for the reviewer whether they would like to raise their score.

---

### Decision · Program_Chairs · 2026-01-26

Accept (Poster)